# IFADAPTER: INSTANCE FEATURE CONTROL FOR GROUNDED TEXT-TO-IMAGE GENERATION

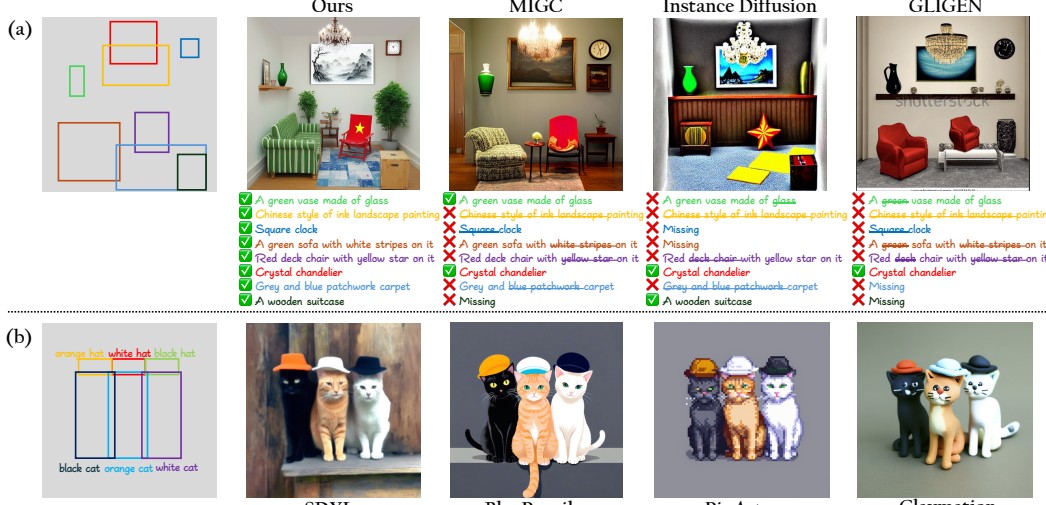

Figure 1: We present **IFAdapter**, a novel approach designed to exert fine-grained control over localized content generation in pretrained diffusion models. (a) IFAdapter has the capacity to generate intricate features with precision. (b) The plug-and-play design of IFAdapter enables it to be seamlessly applied to various community models.

## ABSTRACT

While Text-to-Image (T2I) diffusion models excel at generating visually appealing images of individual instances, they struggle to accurately position and control the features generation of multiple instances. The Layout-to-Image (L2I) task was introduced to address the positioning challenges by incorporating bounding boxes as spatial control signals, but it still falls short in generating precise instance features. To address this Instance Feature Generation (IFG) task, we introduce the Instance Feature Adapter (IFAdapter). The IFAdapter enhances feature depiction by incorporating additional appearance tokens and utilizing an Instance Semantic Map to align instance-level features with spatial locations. The IFAdapter guides the diffusion process as a plug-and-play module, making it adaptable to various community models. For evaluation, we contribute an IFG benchmark and develop a verification pipeline to objectively compare models' abilities to generate instances with accurate positioning and features. Experimental results demonstrate that IFAdapter outperforms other models in both quantitative and qualitative evaluations.

## 1 INTRODUCTION

The advent of diffusion models has revolutionized the field of Text-to-Image (T2I) synthesis (Ho et al., 2020; Podell et al., 2023; Baldridge et al., 2024; Betker et al., 2023; Rombach et al., 2022; Yang et al., 2023a). Despite their exceptional performance in generating high-quality images of single objects, these models remain limited in composing multiple objects into an exquisite image. There are two key challenges underscore this limitation: 1) The inability of natural language in conveying precise spatial information impedes expression of user intent to the model, resulting in

poor image composition in the generated images. 2) Relying solely on a given text prompt describing the attributes of multiple objects, existing models often fails to bind the detailed features to the correct object instances (Feng et al.).

Recent advancements in the Layout-to-Image (L2I) task (Li et al., 2023; Wang et al., 2024c; Zhou et al., 2024b; Kim et al., 2023; Bar-Tal et al., 2023) have partially mitigated such limitation and achieved precise instance-level position control by incorporating bounding boxes as spatial signals. However, in terms of instance feature generation, most state-of-the-art (SoTA) L2I methods can only accurately depict coarse features of an instance (e.g., color attribution), while struggling to generate more complex, fine-grained features. This shortcoming limits the models' applicability in scenarios such as graphic design and art design, where local high-grade details are essential. To simultaneously track the improvement of layout accuracy and feature generation accuracy, a more challenging task, termed Instance Feature Generation (IFG) task, is proposed by InstanceDiffusion. But We found that existing T2I methods, including InstanceDiffusion do not perform satisfactorily on the IFG task, as shown in Figure 1(a). Upon experiment and analysis, we attribute this underperformance to two restrictions: 1) Insufficient detailed descriptions: Most L2I methods rely solely on category labels as descriptions for instances during training. This approach causes samples with detailed descriptions to become out-of-distribution during inference. 2) Insufficient feature information: Existing designs mostly use a single contextualized token to guide the feature generation of each instance. Although this token effectively captures the coarse semantics of the instance (Chen et al., 2024), it is limited in generating high-frequency appearance features.

In this work, we propose the **I**nstance **F**eature **Adapter** (IFAdapter) to address the aforementioned restrictions. First, to address issues related to the training data, we utilize existing SoTA Vision-Language Models (VLMs) for annotation, generating a dataset with detailed instance-level descriptions. Subsequently, we implement two meticulously designed components to address the challenges of instance positioning and feature representation. 1) Appearance Tokens: To address the loss of detailed feature information in instances, the IFAdapter introduces novel learnable appearance queries. These queries extract instance-specific feature information from descriptions, forming appearance tokens that work alongside $EoT$ tokens, thereby enabling more precise control over the generation of instance features; 2) Instance Semantic Map: In contrast to sequence-to-2D grounding conditions (Li et al., 2023; Wang et al., 2024c), IFAdapter constructs a 2D semantic map to correlate instance features with designated spatial locations. This map-like condition provides enhanced spatial guidance, reinforcing the spatial prior and preventing the leakage of instance features. In regions where multiple instances overlap, a gated semantic fusion mechanism is employed to resolve feature confusion. The IFAdapter integrates the semantic map only within a subset of cross-attention layers (Vaswani, 2017) in the diffusion model. This loose coupling allows the IFAdapter to function as a plug-and-play component, enabling its instance-level control capabilities to be transferred across various community models without requiring retraining, as illustrated in 1(b).

For evaluation, previous L2I benchmarks have primarily focused on instance positional accuracy, overlooking instance feature accuracy, which limits their ability to fully assess model performance on the IFG task. To address this limitation, we introduce the COCO-IFG benchmark, designed to evaluate models based on both positional accuracy and precise instance feature generation. Additionally, to overcome the limitations of existing object detection methods, which are incapable of detecting instance features, we integrate SoTA VLMs to facilitate instance feature detection, establishing an objective verification pipeline. Comprehensive experiments on the benchmark demonstrate that IFAdapter significantly enhances instance feature generation accuracy while maintaining precise positional accuracy.

The contributions of this work are as follows:

1. We propose IFAdapter, which utilizes novel appearance tokens and instance semantic map to enhance diffusion models' depiction of instances, enabling high-fidelity instance feature generation.

2. We introduce the COCO IFG benchmark and verification pipeline to evaluate and compare models' performance in grounded instance feature generation.

3. Comprehensive experiments demonstrate that our model outperforms the baselines in both quantitative and qualitative evaluations.

4. The IFAdapter is designed as a plug-and-play component, enabling it to seamlessly empower various community models with layout control capabilities.

## 2 RELATED WORK

**Controllable Diffusion Models** The emergence of diffusion models has significantly propelled advancements in the field of image generation. Controllable Diffusion Models utilize a wide variety of control conditions to generate images with specific content, leading to a proliferation of applications. Semantic control enables precise manipulation of image attributes or features in the generation process by referencing text (Rombach et al., 2022; Saharia et al., 2022b; Ramesh et al., 2022; Chen et al., 2023a) or images (Tang et al., 2023; Saharia et al., 2022a). Spatial control provides fine-grained control over the content in specific regions, such as segmentation-guided (Bar-Tal et al., 2023; Couairon et al., 2023; Wu et al., 2024a), sketch-guided (Voynov et al., 2023), and depth-guided methods (Kim et al., 2022). Recent efforts have concentrated on integrating these spatial control conditions into a unified framework for text-to-image generation, including approaches such as ControlNet (Zhang et al., 2023; Zhao et al., 2024), Composer (Huang et al., 2023), and Adapter-based (Mou et al., 2024) methods. ID and style control emphasize maintaining the consistency of user-specified identity or style in generated images, tuning-based methods guide diffusion models to generate the specified content by fine-tuning (Hu et al., 2021; Ruiz et al., 2023), while tuning-free methods (Ye et al., 2023; Huang et al., 2024; Wang et al., 2024b; Li et al., 2024; Hertz et al., 2024; Wang et al., 2024a) injecting coded condition embedding in the denoising process.

**Layout-to-Image Generation** In the early stages, Layout-to-Image (Layout-to-Image) works primarily hinged on Generative Adversarial Networks (GANs) (Sun & Wu, 2019; 2021; Li et al., 2021; He et al., 2021; Wang et al., 2022; Sylvain et al., 2021). Novel modules and techniques have been proposed to address specific challenges in existing methods, such as object-to-object relations (He et al., 2021; Sylvain et al., 2021), object appearance (Sun & Wu, 2021; He et al., 2021), and handling interactions between bounding boxes (Sylvain et al., 2021; Li et al., 2021; Wang et al., 2022). Nevertheless, withthe rising tide of diffusion-based methods in the generative field, incorporating diffusion techniques into Layout-to-Image methods has led to significant improvements in the quality, diversity, and controllability of generated images. In some earlier works (Cheng et al., 2023; Zheng et al., 2023), semantic control was primarily achieved through the use of entity classes. Some training-free methods (Xiao et al., 2023; Xie et al., 2023; Chen et al., 2024) leverage the prior knowledge of the pre-trained model's semantic control to guide object placement within specific regions. Other approaches (Wang et al., 2024c; Zhou et al., 2024b;a; Li et al., 2023; Yang et al., 2023b; Avrahami et al., 2023) encode layout locations and semantic descriptions into features that are processed by attention mechanisms. The aforementioned methods generally rely on class tags or simple attributes. In contrast, our method employs detailed instance-level descriptions, combined with an adapter design, result in superior performance.

## 3 APPROACH

### 3.1 PRELIMINARIES

**Diffusion Models**. Our method is applied over a pretrained T2I diffusion model, more specifically, a T2I latent diffusion model (LDM) (Rombach et al., 2022). The generation process of the LDM can be regarded as stepwise denoising from a initial Gaussian noise $z \sim \mathcal{N}(0, I)$, conditioned on a textual prompt $y$. The training objective is to minimize the following LDM loss:

$$\mathcal{L}_{LDM} = \mathbb{E}_{z \sim \mathcal{N}(0,I),y,t}[||\epsilon - \epsilon_\theta(z_t, t, E(y))||_2^2], \tag{1}$$

where the $\epsilon_\theta$ is parameterized as a UNet (Ronneberger et al., 2015) and $t$ is the denoising timestep. $E$ is a pretrained text encoder, used to encode $y$ into text embeddings.

**Cross Attention**. In the LDM, text embeddings guide the direction of generation via cross attention operations (Vaswani, 2017), which can be represented using the following equation:

$$\text{Attention}(\mathbf{Q}, \mathbf{K}, \mathbf{V}, \mathbf{M}) = \text{Softmax}(\frac{\mathbf{Q}\mathbf{K}^\top}{\sqrt{d}} + \mathbf{M})\mathbf{V}. \tag{2}$$

The $Q$ is obtained by projecting the image latent code through a Multi-Layer Perceptron (MLP), while $K$ and $V$ are similarly derived from text embeddings. $\mathbf{M}$ is a mask used to adjust attention scores, and $d$ represents the dimensionality of the hidden vector, which helps stabilize the training process.

## 3.2 PROBLEM DEFINITION

In the Instance Feature Generation task, the LDM requires additional conditioning on a set of local descriptors $c = \{(\mathbf{r}_1, \mathbf{l}_1), \ldots, (\mathbf{r}_n, \mathbf{l}_n)\}$. $r_i$ represents the designated generation position for the $i$-th instance, in $[x, y, w, h]$ form. $l_i$ is the corresponding phrase that describes the features of the $i$-th instance. Our method differs from others in that $l_i$ incorporates detailed, extended descriptions of the instance, including aspects such as mixed colors, complex textures, etc. With $c$ serving as auxiliary conditions, the LDM should be able to generate instances with high fidelity in both position and features.

## 3.3 IFADAPTER

In this work, the IFAdapter is designed to control the generation of instance position and features. We employ the open-source Stable Diffusion (SDXL) Podell et al. (2023) as the base model. To address the issue of instance feature loss, we introduce appearance tokens as a supplement to the high-frequency information, as discussed in Sec. 3.3.1. Furthermore, to incorporate a stronger spatial prior for more accurate control over position and features, we use appearance tokens to construct an instance semantic map that guides the generation process, as elaborated in Sec. 3.3.2.

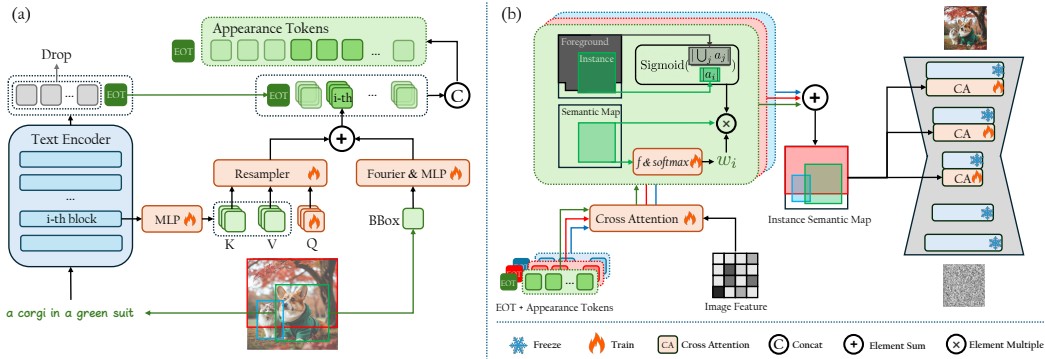

Figure 2: **Structure of proposed IFAdapter.** In (a), we illustrate the generation process of Appearance Tokens. For simplicity, we use the generation process of one instance (the corgi) as example. In (b), we present the construction process of the Instance Semantic Map.

## 3.3.1 APPEARANCE TOKENS

L2I SD enables the generation of grounded instances by incorporating local descriptions and location as additional conditions. Existing approaches (Li et al., 2023; Zhou et al., 2024b; Wang et al., 2024c) typically utilize the contextualized token (the End of Text, $EoT$ token) produced by the pretrained CLIP text encoder (Radford et al., 2021) to guide the generation of instance features. Although the $EoT$ token plays a crucial role in foreground generation, it primarily focuses on generating coarse structural content (Wu et al., 2024b; Chen et al., 2024) and requires additional tokens to complement high-frequency details. As a result, existing L2I methods that discard all other tokens are unable to generate detailed instance features. One naive mitigation approach would be to use all tokens (77 in total) generated by the CLIP text encoder as instance-level conditions. However, this approach would significantly increase the computational burden during both training and inference. Moreover, these 77 tokens include a substantial number of padding tokens that do not contribute to the generation. While removing padding tokens can reduce computational costs, this strategy is incompatible with batch training due to the varying lengths of the descriptions. To address this,

we propose compressing the feature information into a small set of appearance tokens and utilizing these tokens to complement the $EoT$ token.

Drawing inspiration from the Perceiver design (Ye et al., 2023; Alayrac et al., 2022), we employ a set of learnable appearance queries to interact with instance description embeddings through cross attention, thereby extracting text feature and forming appearance tokens, as shown in Fig. 2 (a). It is worth noting that the appearance queries only interact with word tokens, thus converting descriptions of arbitrary length into fixed-length appearance tokens. In addition, to obtain text features of different entangled granularities, the query tokens also interact with the shallower layers of the text encoder. By combining the appearance tokens with location embeddings, $\mathbf{h}^l \in \mathbb{R}^{L \times d}$ are obtained from the text encoder layer $l$. The $L$ denotes the number of appearance tokens. This process can be expressed using the following formula:

$$\mathbf{H} = [\mathbf{h^{l_1}}, \ldots, \mathbf{h^{l_k}}]$$
$$\text{where} \quad \mathbf{h^l} = \text{Resampler}(\mathbf{Q_a}, \mathbf{K^l}, \mathbf{V^l}) + \text{MLP}(\text{Fourier}(\boldsymbol{r})). \tag{3}$$

For the sake of clarity, we use the generation of appearance tokens for a single instance as an example. The Resampler is adapted from Perceiver, composed of multiple transformer blocks. $Q_a$ represents the appearance queries, while $\mathbf{K}^l$ and $\mathbf{V}^l$ are obtained by projecting the text features extracted from the $l$-th layer of the text encoder. The Fourier is the Fourier embedding (Mildenhall et al., 2021), combined with a MLP to project $r$ to the feature space. Finally, the appearance tokens at $k$ different granularities are concatenated into $\mathbf{H} \in \mathbb{R}^{(kL) \times d}$ to serve as the generation guidance for each instance.

### 3.3.2 INSTANCE SEMANTIC MAP-GUIDED GENERATION

Along with ensuring the generation of detailed instance features, the IFG task also requires instances to be generated at designated locations. Previous method (Li et al., 2023) uses sequential grounding tokens as conditions, which lack robust spatial correspondence, potentially leading to issues such as feature misplacement and leakage. Therefore, in our work, we introduce a map called the Instance Semantic Map (ISM) as a stronger guiding signal. Since the generation of all instances is guided by the ISM, two major considerations must be addressed when constructing the map: (1) generating detailed and accurate features for each instance while avoiding feature leakage, and (2) managing overlapping regions where multiple instances are present. To address these concerns, we first generate each instance in isolation and then aggregate them in the overlapping regions. The following sections will provide a detailed explanation of these processes.

**Per-instance Feature Generation**. Avoiding interference from extraneous features is crucial for the precise generation of high-quality instance details. To achieve this objective, we first generate the semantic map of each instance individually. Specifically, for the $i$-th instance, we transform its corresponding location $\mathbf{r}_i$ into the following mask $\mathbf{m}_i$:

$$\mathbf{m}_i(x,y) = \begin{cases} 0 & \text{if } [x,y] \in \mathcal{R}_i \\ -\infty & \text{if } [x,y] \notin \mathcal{R}_i \end{cases}, \tag{4}$$

where $\mathcal{R}_i$ represents the coordinates within the region indicated by $\mathbf{r}_i$. By employing Eq. 2, we can obtain the semantic map $s_i$ for the $i$-th instance:

$$s_i = \text{Attention}(\mathbf{Q}, \mathbf{K_i}, \mathbf{V_i}, \mathbf{m_i}), \tag{5}$$

where $K_i$ and $V_i$ are projected from the concatenation of the appearance tokens $\mathbf{H}$ and $EoT$ token of $i$-th instance, the $\mathbf{Q}$ is derived from the image latent code.

**Gated Semantic Fusion**. After obtaining the semantic maps for each instance, the next step is to blend these maps to derive the final ISM, as shown in Fig. 2 (b). A critical issue during the map integration process is how to handle the latent pixels that are associated with multiple instances. Previous method (Jia et al., 2024) average the representations from multiple instances. While this approach is simple, it may lead to feature conflicts between different instances. Intuitively, the visual features in regions where multiple instances overlap should be dominated by the instance closest to the observer (i.e., the one with the smallest depth). Therefore, the weights of different instances in

overlapping regions should vary. For clarity, we use the integration process at pixel location $(x, y)$ as an example. The representations of each instance are first projected into a scalar representing importance through a trainable lightweight network $f$. Then, the Softmax operation normalizes the importance across different instances, yielding their respective weights. This process can be described by the following equation:

$$[w_1(x, y), \ldots, w_n(x, y)] = \text{Softmax}(f(s_1(x, y)), \ldots, f(s_n(x, y))), \quad (6)$$

where $w_i(x, y)$ denotes the weight of instance $i$ at location $(x, y)$.

In addition to the instance representation, the size of the instance also influences its weight. This design is motivated by the following consideration: when the region of a small instance is completely covered by a larger instance, it is necessary to prevent the smaller instance from being "assimilated". Therefore, the proportion of the area occupied by the instance in the foreground is also considered, with smaller instance being assigned greater weight. Using the instance representations and their respective weights, the final representation for a latent pixel position $(x, y)$ is obtained using the following formula:

$$\mathbf{D}(x, y) = \sum_i w_i(x, y) \cdot \text{Sigmoid}(\frac{|\bigcup_j^n a_j|}{|a_i|}) \cdot s_i(x, y), \quad (7)$$

$a_i$ represents the area occupied by instance $i$. After the aforementioned steps, the ISM is constructed. Finally, ISM interacts through the following duplicate cross attention layers (Ye et al., 2023) to guide the generation of salient regions:

$$\text{Attn} = \text{Attention}(\mathbf{Q}, \mathbf{K}, \mathbf{V}, 0) + \tanh(\lambda) \cdot (1 - \mathcal{M}_{bg}) \odot \mathbf{D}, \quad (8)$$

where $\mathcal{M}_{bg}$ is a binary mask with the background area set to 1, and $\lambda$ is a trainable parameter initialized to 0 to prevent pattern collapse during the initial training phase.

## 3.4 LEARNING PROCEDURE

During training, we freeze the parameters of the SD, training only the IFAdapter. The loss function used for training is the LDM loss with instance-level condition incorporated:

$$\mathcal{L}_{IFA} = \mathbb{E}_{z \sim \mathcal{N}(0,I), y, t}[||\epsilon - \epsilon_\theta(z_t, t, E(y)), c||_2^2] \quad (9)$$

To enable our method to perform classifier-free guidance (CFG) (Ho & Salimans, 2022) during the inference phase, we randomly set the global condition $y$ and local condition $c$ to 0 during training.

## 4 EXPERIMENTS

### 4.1 IMPLEMENTATION DETAILS

We described the basic setup for training our model. For more details, please refer to the appendix.

**Training dataset.** We use the COCO2014 (Lin et al., 2014) dataset and a 1 million subset from LAION 5B (Schuhmann et al., 2022) as our data sources. Following previous methods (Wang et al., 2024c; Zhou et al., 2024b), we utilize Grounding-DINO (Liu et al., 2023) and RAM (Zhang et al., 2024) to annotate the instance positions within the images. We then employ the state-of-the-art visual language models (VLMs) QWen (Bai et al., 2023) and InternVL (Chen et al., 2023b) to generate captions for the images and individual instance.

**Training details.** We use SDXL (Podell et al., 2023), known for its strong detail generation capabilities, as our base model. The IFAdapter is applied to a subset of SDXL's mid-layers and decoder layers, which significantly contribute to foreground generation. We trained the IFAdapter using the AdamW (Loshchilov et al., 2017) optimizer with a learning rate of 0.0001 for 100,000 steps and a batch size of 160. During training, there was a 15% chance of dropping the local description and a 30% chance of dropping the global caption. For inference, we used the EulerDiscreteScheduler (Karras et al., 2022) with 30 sample steps and set the classifier-free guidance (CFG) scale to 7.5.

## 4.2 Experimental Setup

**Baselines.** We compared our approach with previous SoTA L2I methods, including training-based methods InstanceDiffusion (Wang et al., 2024c), MIGC (Zhou et al., 2024b), and GLIGEN (Li et al., 2023), as well as the training-free methods DenseDiffusion (Kim et al., 2023) and MultiDiffusion (Bar-Tal et al., 2023).

**Evaluation dataset.** Following the previous setup (Li et al., 2023; Zhou et al., 2024b; Wang et al., 2024c), we constructed the COCO IFG benchmark on the standard COCO2014 dataset. Specifically, we annotate the locations and local descriptions in the validation set using the same approach as in the training data. Each method is required to generate 1,000 images for validation.

**Evaluation Metrics.** For the validation of the IFG task, it is imperative that the model generates instances with accurate features at the appropriate locations.

- **Instance Feature Success Rate.** To verify spatial accuracy and description-instance consistency, we propose the Instance Feature Success (IFS) rate. The calculation of the IFS rate involves two steps. **Step 1, Spatial accuracy verification:** We begin by using Grounding-DINO to detect the positions of each instance. Next, we compute the Intersection over Union (IoU) between the detected positions and the Ground Truth (GT) positions, selecting the GT with the highest IoU as the corresponding match for that instance. If the highest IoU is less than 0.5, the instance generation is considered **unsuccessful**. **Step 2, Local feature accuracy verification:** Previous methods (Avrahami et al., 2023; Zhou et al., 2024b) primarily employ local CLIP for verifying local features. However, CLIP focuses on overall semantics and is not well-suited for capturing fine visual details (Yuksekgonul et al., 2023). Therefore, we utilize VLMs in conjunction with the prompt engineering technique to achieve more precise verification of local details. For each local region identified in Step 1, we prompt the VLMs to determine whether the content within the cropped region aligns with the corresponding description. If the VLM confirms that the content matches the prompt, the instance is marked as successful. The Instance Foreground Success (IFS) rate is then calculated as the ratio of successful instances to the total number of instances. Additionally, we report the Grounding-DINO Average Precision (AP) score to independently validate the positional accuracy of instance location generation.

- **Fréchet Inception Distance (FID).** FID (Heusel et al., 2017) measures image quality by calculating the feature similarity between generated and real images. We compute the FID using the validation set of COCO2017.

- **Global CLIP Score.** The global caption of the image primarily describes the overall semantics of the image. Therefore, we use the CLIP score to evaluate Image-Caption Consistency.

## 4.3 Comparison

### 4.3.1 Quantitative analysis.

Tab. 1 presents our qualitative results on the IFG benchmark, including metrics of IFS Rate, Spatial accuracy, and the Image Quality.

**IFS Rate.** To calculate the IFS rate, we utilize three state-of-the-art (SoTA) vision-language models (VLMs): QWenVL (Bai et al., 2023), InternVL (Chen et al., 2023b), and CogVL (Wang et al., 2023). This multi-model approach ensures a more comprehensive and rigorous validation. As shown in Tab 1, our model outperforms the baseline models in all three IFS rate metrics. The introduction of appearance tokens and the incorporation of dense instance descriptions in training have significantly enhanced our model's ability to generate accurate instance details. It is worth noting that InstanceDiffusion achieves a higher IFS rate compared to other baselines. This is likely due to its training dataset also contains dense instance-level descriptions. This observation further underscores the necessity of high-quality instance-level annotations.

**Spatial Accuracy.** As can be observed from Tab 1, IFAdapter achieves the best results in Grounding-DINO AP. This success can be attributed to our map-guided generation design, which incorporates additional spatial priors, leading to more accurate generation of instance locations.

| Methods | IFS Rate(%) | | | Spatial(%) | Quality | |
|---|---|---|---|---|---|---|
| | QwenVL ↑ | InternVL ↑ | CogVL ↑ | AP ↑ | CLIP ↑ | FID ↓ |
| Real images | 92.8 | 82.2 | 69.9 | 75.3 | - | - |
| InstanceDiffusion | 69.6 | 49.7 | 38.2 | 43.1 | 23.3 | 26.8 |
| GLIGEN | 44.8 | 25.8 | 17.5 | 18.4 | 23.5 | 29.7 |
| MIGC | 62.8 | 40.7 | 27.5 | 32.5 | 22.9 | 26.0 |
| MultiDiffuion | 58.1 | 47.0 | 34.2 | 36.9 | 22.8 | 28.3 |
| DenseDiffusion | 38.7 | 26.0 | 19.7 | 22.2 | 20.1 | 29.9 |
| **Ours** | **79.7** | **68.6** | **61.0** | **49.0** | **25.1** | **22.0** |

Table 1: **Evaluation on COCO IFG benchmark.** To perform a more rigorous and comprehensive experiment for calculating the IFS rate, we utilize three different VLMs. For spatial accuracy, we report the Grounding-DINO AP. To assess overall image quality, we measure the CLIP score and FID. The ↑ indicates that a higher value is better, while ↓ signifies the opposite.

**Image Quality.** As shown in Table 1, our method demonstrates a higher CLIP Score, indicating that enhancing local details contributes to the simultaneous improvement of image-caption consistency. Additionally, our method achieves a lower FID, suggesting that the images generated by our approach are of higher quality compared to the baselines. We attribute this improvement to the adapter-like design of our model, which enables spatial control without significantly compromising image quality.

### 4.3.2 QUALITATIVE ANALYSIS.

In Fig. 1(a), we present generation results for a scene with multiple complex instances. We further evaluate the models' ability to generate instances with diverse features in Fig. 3. As shown, our method demonstrates the highest level of fidelity across various types of instance details.

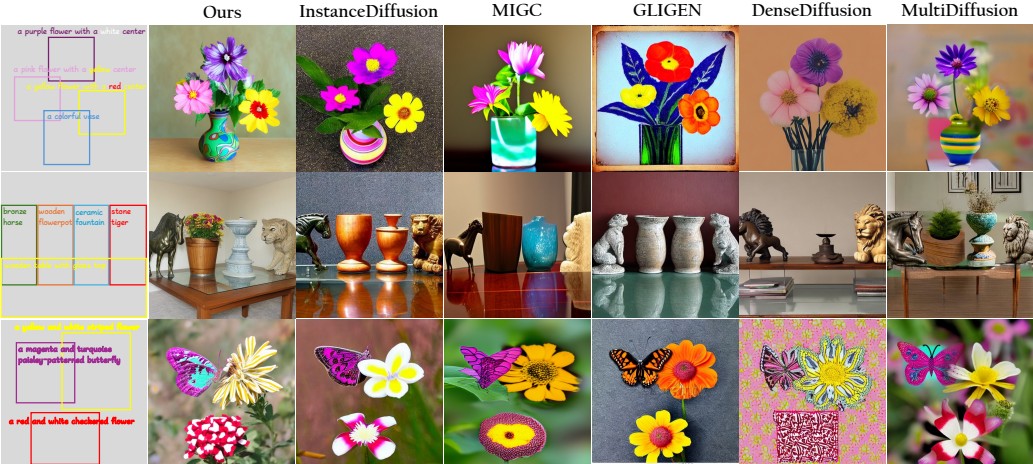

Figure 3: **Qualitative results.** We compare the models' ability to generate instances with different types of features, including mixed colors, varied materials, and intricate textures.

### 4.4 USER STUDY.

Although VLMs can verify instance details to a certain extent, a gap remains compared to human perception. Therefore, we invited professional annotators for further validation.

| Methods | Spatial | | Instance Details | | Aesthetics | |
|---------|---------|---------|---------|---------|---------|---------|
| | Score ↑ | Pref. Rate ↑ | Score ↑ | Pref. Rate ↑ | Score ↑ | Pref. Rate ↑ |
| InstanceDiffusion | 4.44 | 44.4% | 3.82 | 33.3% | 2.99 | 14.8% |
| GLIGEN | 3.96 | 14.2% | 2.54 | 3.7% | 2.44 | 3.7% |
| MIGC | 4.30 | 33.3% | 3.39 | 7.4% | 2.54 | 3.7% |
| **Ours** | **4.85** | **88.9%** | **4.69** | **88.9%** | **4.10** | **96.2%** |

Table 2: **Results of user study.** We conducted a user study to evaluate the spatial generation accuracy, instance detail generation effectiveness, and aesthetic index of the L2I methods. Evaluators were provided with the image layout and the corresponding image, and they were asked to rate the aforementioned three dimensions on a scale of 0 to 5. A score of 0 represents the lowest rating, while 5 represents the highest rating. We also reported the user preference rate (Pref. Rate), which represents the proportion of the highest scores obtained by the methods.

**Setup.** We conducted a study comprising 270 questions, each associated with a randomly sampled generated image. Evaluators were asked to rate image quality, instance location accuracy, and instance details. In total, 30 valid responses were collected, yielding 7,290 ratings.

**Results.** As seen in Tab. 2, our method achieves the highest scores and user preference rate across all three dimensions. Notably, the trends in these dimensions are consistent with those in Table 1, further demonstrating the effectiveness of VLM validation.

## 4.5 INTEGRATION WITH COMMUNITY MODELS

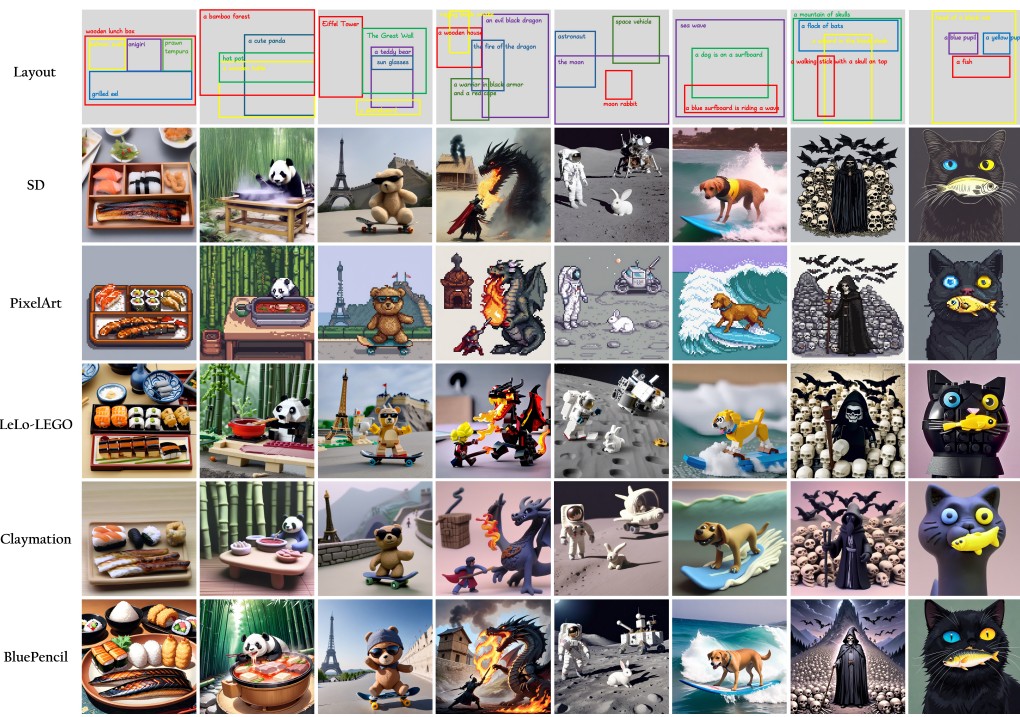

Figure 4: The IFAdapter can seamlessly integrate with community diffusion models.

Thanks to the plug-and-play design of the IFAdapter, it can impose spatial control on pretrained diffusion models without significantly compromising the style or quality of the generated images. This capability enables the IFAdapter to be effectively integrated with various community diffusion models and LoRAs (Hu et al., 2021). As illustrated in Fig. 4, we applied IFAdapter to several community

models, including PixlArt (NeriJS, 2023), LeLo-LEGO (LordJia, 2024), Claymation (DoctorDiffusion, 2024), and BluePencil (blue_pen5805, 2024). The generated images not only adhere to the specified layouts but also accurately reflect the respective styles.

## 4.6 ABLATION STUDY

Figure 5: **Qualitative results of variants of IFAdapter.**

This ablation study primarily explores the roles of appearance tokens and $EoT$ token in instance generation. The results of the ablation experiments are presented in Tab. 3.

**appearance tokens.** The removal of appearance tokens leads to a decrease in the model's IFS rate and FID, indicating a loss of detailed features. Furthermore, as illustrated in Fig. 5, the images generated without appearance tokens exhibit instance feature mismatches, further demonstrating that appearance tokens are primarily responsible for generating high-frequency appearance features.

$EoT$ **token.** The IFS rate significantly decreases when generating without the $EoT$ token. This is primarily because the $EoT$ token is responsible for generating the coarse semantics of instances. Additionally, Fig. 5 indicates that removing the $EoT$ token results in semantic-level issues, such as instance category errors and instance omissions.

If the $EoT$ token and appearance tokens are both removed, the model reverts to the baseline text-to-image diffusion. Consequently, it lacks the capability for instance-level generation, resulting in poor performance on IFG task.

| appearance tokens | EoT token | IFS Rate(%) | | | Spatial(%) | Quality | |
| --- | --- | --- | --- | --- | --- | --- | --- |
| | | QwenVL ↑ | InternVL ↑ | CogVL ↑ | AP ↑ | CLIP ↑ | FID ↓ |
| | | 17.3 | 9.5 | 7.4 | 9.3 | 23.7 | 30.2 |
| | ✓ | 69.6 | 63.9 | 53.5 | 45.9 | 24.1 | 27.2 |
| ✓ | | 29.9 | 16.2 | 12.0 | 12.3 | 24.3 | 44.7 |
| ✓ | ✓ | **79.7** | **68.6** | **61.0** | **49.0** | **25.1** | **22.0** |

Table 3: **Quantitative results of variants of IFAdapter.**

## 5 CONCLUSION

In this work, we propose IFAdapter to exert fine-grained, instance-level control on pretrained Stable Diffusion models. We enhance the model's ability to generate detailed instance features by introducing Appearance Tokens. By utilizing Appearance Tokens to construct an instance semantic map, we align instance-level features with spatial locations, thereby achieving robust spatial control. Both qualitative and quantitative results demonstrate that our method excels in generating detailed instance features. Furthermore, due to its plug-and-play nature, IFAdapter can be seamlessly integrated with community models as a plugin without the need for retraining.

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
