# APPENDIX FOR IFADAPTER: INSTANCE FEATURE CONTROL FOR GROUNDED TEXT-TO-IMAGE GENERATION

## A  MORE IMPLEMENTATION DETAILS

**Dataset Construction.** We utilized VLMs to construct a dataset containing detailed instance-level descriptions for model training and validation. The data construction pipeline can be broadly divided into three stages and is illustrated in Fig. 6.

1. **Open-set object detection:** In the first stage, we used Grounding DINO (Liu et al., 2023) in combination with RAM (Zhang et al., 2024) to label instances in images based on COCO-provided labels. This approach helps mitigate the issue of missing objects in the original COCO dataset annotations.

2. **Bounding box post-processing:** After obtaining the instances and their corresponding bounding boxes, we applied the Non-Maximum Suppression (NMS) algorithm to eliminate duplicate bounding boxes and discarded small objects with an area of less than 0.01.

3. **Annotating instance appearance:** Based on the detected bounding boxes, we cropped each instance from the image and utilized Qwen (Bai et al., 2023) to generate detailed descriptions.The prompt provided to the model is: "You are given a cropped image. Please provide a clear and concise description in one sentence, emphasizing the primary object, including its appearance and posture."

As shown in the rightmost part of Fig. 6, after being processed through our pipeline, the images are annotated with both the positions and feature labels of instances, which are then used for subsequent training and evaluation.

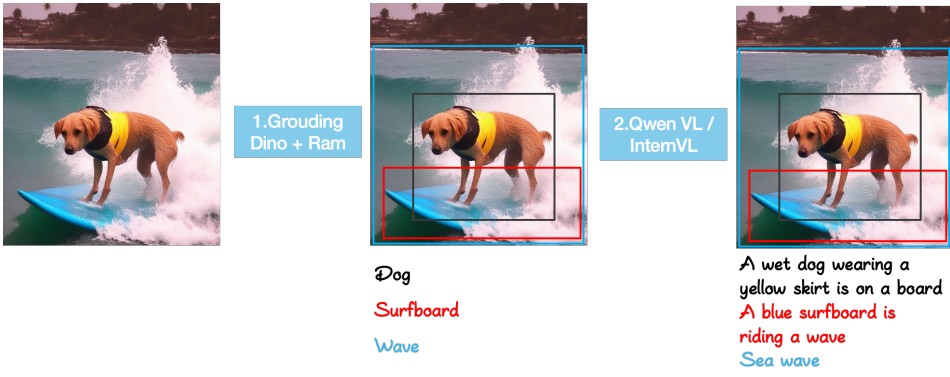

Figure 6: **The pipeline of dataset construction.**

**Training Details.** We trained the model using 16 Ascend 910B NPUs for 8 days. To enable batch training, we fixed the number of instances per sample at 8. For samples containing more than 8 instances, we randomly discarded the excess instances. For those with fewer than 8 instances, we padded the instance features with zero vectors and padded the bounding boxes with $[0, 0, 0, 0]$. Inspired by MIGC Zhou et al. (2024b), we implement $f$ in Eq. 6 using CBAM Woo et al. (2018). Other networks like SENet Hu et al. (2019) could also be suitable.

**IFAdapter Integration.** The main architecture of SDXL is a UNet (Ronneberger et al., 2015), consisting of mid-layers, upper layers, and down layers.The UNet (Ronneberger et al., 2015) is composed of attention blocks, where each attention block consists of ten transformer blocks. Exerting control over all attention blocks could potentially degrade SDXL's original performance and introduce a significant computational burden. Therefore, in our approach, the IFAdapter is applied to a subset of the layers in SDXL (Podell et al., 2023). More specifically, IFAdapter applies control to the mid-layers and the first two attention blocks in the upper layers. Within each attention block, control is exerted over the first five transformer blocks. This design is guided by the visualiza-

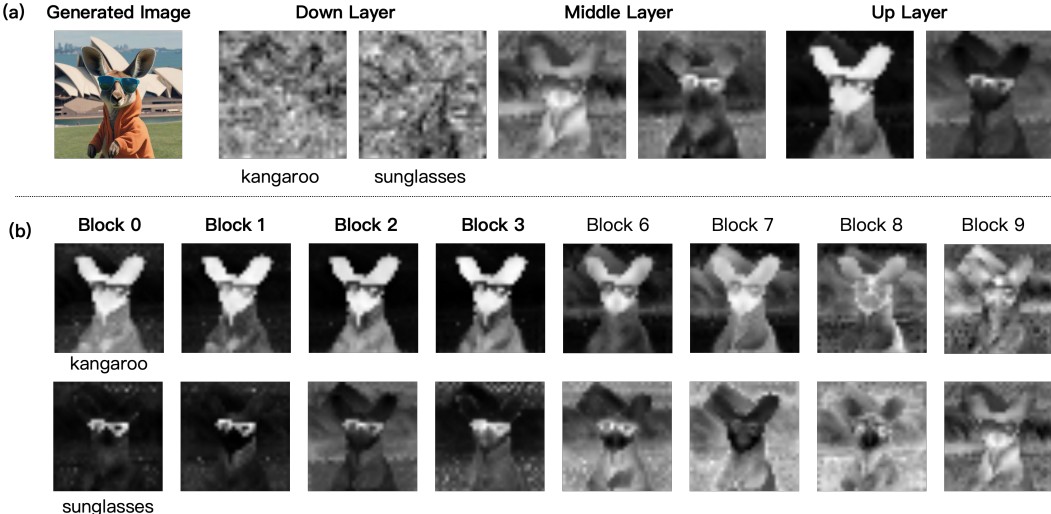

Figure 7: **Visualization of the cross attention maps.** (a) Cross-attention maps from the upper, mid, and lower layers of SDXL. (b) Cross-attention maps from different transformer blocks within the mid-layer of SDXL.

tion technique introduced in previous research (Chen et al., 2024), which helps identify transformer blocks that are crucial for the generation of the foreground.

As illustrated in Fig. 7 (a), compared to the down layers, the cross-attention maps in the mid-layers and upper layers exhibit higher values concentrated on the instances, indicating a stronger emphasis on foreground elements. In Fig. 7 (b), it is observed that as the depth of the transformer blocks increases, the focus of the cross-attention maps gradually shifts from the instances to the broader image context. This implies that the shallow transformer blocks contribute more significantly to instance generation. Since the IFAdapter requires precise control over instances, we specifically apply control to the shallow blocks in the mid-layers and upper layers.

## B  EVALUATION DETAILS

**Baseline Settings.** We compared our approach with previous SoTA L2I methods, including training-based methods InstanceDiffusion (Wang et al., 2024c), MIGC (Zhou et al., 2024b), and GLIGEN (Li et al., 2023), as well as the training-free methods DenseDiffusion (Kim et al., 2023) and MultiDiffusion (Bar-Tal et al., 2023). Since all these methods are compatible with the input format of our dataset, they are executed using the official code and default configurations.

**VLM Evaluation Details.** We employed VLMs in the automatic evaluation pipeline for instance-level detail quality assessment. To ensure a more effective feature evaluation, we employed prompt engineering to guide the context of the VLMs toward focusing on the details of the instances.

First, the VLM is provided with a cropped image of the instance and prompted with the following instruction: `"Please provide a detailed description of the primary object in this image."` After this step, the context of the VLM is focused on the details of the appearance of the instance. Next, we prompt the VLM to determine whether the features of the instance match its corresponding description using the following instruction: `"Please verify whether the appearance of the primary object matches the following description: ${local description}. Your answer must begin with 'yes' or 'no'."`. The ${local description} refers to the description used to generate the instance. The instance is considered correctly generated only if the VLM responds with 'Yes'.

**User Study Details.** In Fig. 8, we provide an example question from our user study.

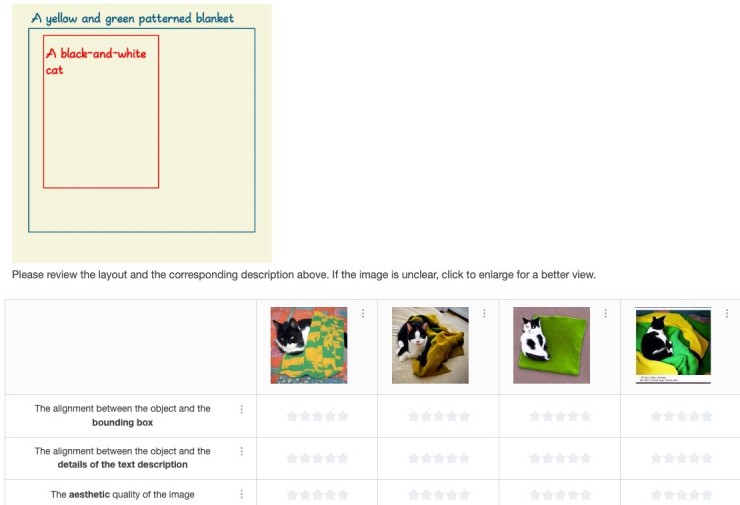

Figure 8: **An example question from the user study.**

Since matching instances with bounding boxes and verifying their features is a demanding task, we limited the number of instances per image to fewer than four to prevent annotator fatigue, which could compromise verification accuracy.

## C   MORE QUANTITATIVE RESULTS

| **Method** | $AP^{box}$ | $AP_{50}^{box}$ | $AP^{box}$ | $AP_L^{box}$ | $AP_s^{box}$ |
|---|---|---|---|---|---|
| InstanceDiffusion | 38.8 | 55.4 | 52.9 | - | - |
| IFAdapter | 29.4 | 52.9 | 39.8 | 42.0 | 6.0 |

Table 4: Evaluation with the setup of InstanceDiffusion.

In addition to the basic quantitative results on the IFG benchmark, we also report the results obtained using the InstanceDiffusion experimental setup. In this setup, the validation set is from the COCO $val$ set. The instance position generation accuracy is evaluated using the $AP$ metrics obtained through YOLO v8 detection. The experimental results are presented in Tab. 4.

The experimental results suggest that our method performs worse than InstanceDiffusion under its experimental setup. As shown in the table, our method achieves lower $AP_s^{box}$ for small objects. After visualizing the samples, we believe this is primarily due to the presence of many small instances in the COCO validation set. Our approach leverages an Instance Semantic Map with a 16×16 resolution as guidance. Consequently, for bounding boxes occupying small areas, positional deviations are relatively large, leading to inaccuracies in the position generation of small instances. In contrast, methods like InstanceDiffusion, which represent objects using tokens, are relatively better for small-object generation.

But we would like to highlight that this shortcoming does not impact IFA's performance under IFG setup. This is because only objects with larger areas typically exhibit noticeable appearance features, and generating small objects is not the primary target of the IFG task.

## D   MORE ABLATION

**Extract appearance tokens from multi-granularities.** In constructing the appearance tokens, we use appearance queries to interact with different layers of the text encoder. We conducted ablation studies to evaluate this design. In the comparison method, appearance tokens are generated solely by interacting with the deepest layer of the text encoder.

| Mode | IFS Rate(%) | | |
|---|---|---|---|
| | QwenVL ↑ | InternVL ↑ | CogVL ↑ |
| Single | 75.6 | 64.8 | 57.4 |
| Multi | **79.7** | **68.6** | **61.0** |

Table 5: Additional ablation study.

As shown in the Tab. 5, using text features from multiple layers improves the quality of instance feature generation. This is because text features are progressively entangled through the attention layers. By interacting with the shallower layers, the appearance tokens can more flexibly integrate text features at varying granularities, thereby enhancing the representation of instance information.

# E ANALYSIS OF GATED SEMANTIC FUSION

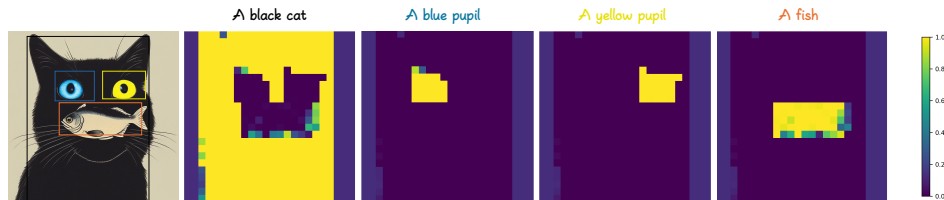

Figure 9: **Visualization of different instance weights in the gated semantic fusion process.**

IFAdapter first generates feature maps for individual instances and then combines them using Gated Semantic Fusion to produce an Instance Semantic Map that integrates the semantics of all instances. During the fusion process, IFAdapter employs a lightweight neural network $f$ to determine the weight of each instance at each latent location, allowing for a weighted blending of the features of the instance.

To provide a more intuitive understanding of the Gated Semantic Fusion process, we visualize the weights assigned to different instances in Fig. 9. As shown, $f$ effectively identifies salient features in the Instance Semantic Map at various positions and assigns greater weights to the corresponding instances, enabling efficient fusion of features in overlapping regions. Notably, $f$ is not explicitly supervised during training; its ability to identify instance importance scores is implicitly learned through Eq. 9.

We hypothesize that this effectiveness stems from the diffusion model features containing a certain degree of depth information Ke et al. (2024), which $f$ learns to map to weights during the training process.

# F ANALYSIS OF COMPUTATIONAL COST IN INFERENCE

| Mode | GPU Memory Usage (MiB) | Inference Latency (Seconds) |
|---|---|---|
| SDXL | 16601 | 5.32 |
| IFA (4 instances) | 17875 | 8.03 |
| IFA (8 instances) | 19485 | 8.77 |
| IFA (16 instances) | 19651 | 10.73 |

Table 6: Memory usage of inference.

The additional computational overhead introduced by the IFA primarily stems from the extraction of appearance tokens, the generation of instance semantic maps, and the Gated Semantic Fusion process. In this section, we analyze the computational burden introduced by each of these components.

1. **Extraction of Appearance Tokens:** IFA uses appearance queries and text embeddings to perform cross-attention, extracting high-frequency feature information. Since this process only needs to be performed once per image generation, it does not introduce significant computational overhead.

2. **Generation of Instance Semantic Maps:** This is the most computationally expensive part of IFA. For each instance, an additional cross-attention operation is performed between the textual features (EoT token + appearance tokens) and the image features, which contributes to the increased computational cost.

3. **Gated Semantic Fusion Process:** Since this process utilizes a lightweight convolutional neural network to perform weighted aggregation of the instance semantic maps, it does not introduce significant computational overhead.

The above analysis indicates that the generation of instance semantic maps contributes the most to the computational overhead, and this overhead increases with the number of instances. To quantify the additional computational burden, we conducted experiments comparing the model's inference memory usage. The results are reported in Fig. 3. The experiment is conducted on one NVIDIA A100 GPU.

The results indicate that, compared to the base model, IFA does not introduce significant computational overhead. We attribute this to the fact that IFA applies control only to a small subset of cross-attention layers, making it relatively lightweight.

In terms of inference speed, the time required to generate Instance Semantic Maps increases as the number of instances grows, leading to a corresponding increase in the overall inference time. As previously analyzed, the bottleneck in inference speed lies in the generation process of Instance Semantic Maps.

## G    MORE QUALITATIVE RESULTS

We provide additional qualitative comparison results between IFAdapter and baseline methods in Fig. 10 and Fig. 11. Specifically, Fig. 10 shows the zero-shot results, while Fig. Fig. 11 presents the results on the IFG benchmarks.

In addition, we present the generation results of our method under densely packed instances, as shown in Fig. 12. It can be observed that even under such a challenging layout condition, our method successfully balances positional accuracy and appearance fidelity for each instance.

## H    LIMITATIONS

We found that the dataset constructed by our pipeline contains some low-quality samples, such as multiple bounding boxes corresponding to the same instance or a single bounding box containing multiple instances. These samples may negatively affect the model performance.

The IFAdapter struggles to generate very small objects because it leverages an Instance Semantic Map for guidance. This leads to relatively large positional deviations of bounding boxes that occupy small areas, hindering the effective generation of very small instances.

When combining the IFAdapter with the community's LoRA for T2I generation, we observed that in some cases, the generated instances exhibited a loss of appearance details. We speculate that this may be due to a conflict between LoRA and the IFAdapter in controlling high-frequency information.

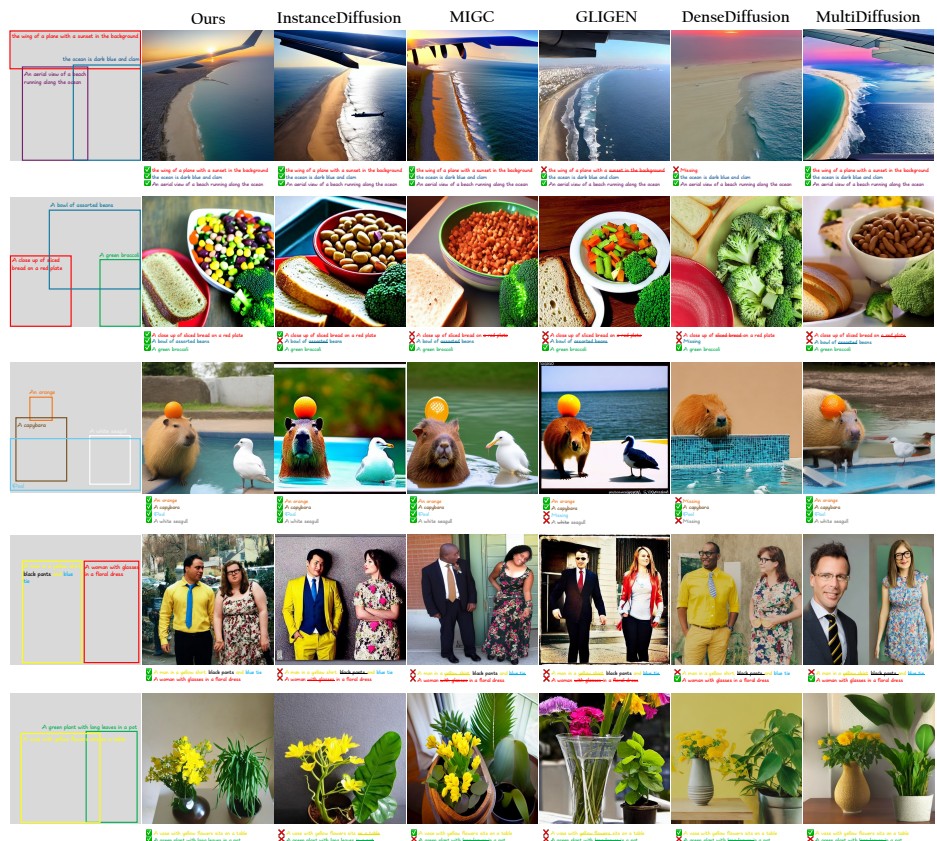

Figure 10: **Additional qualitative results.**

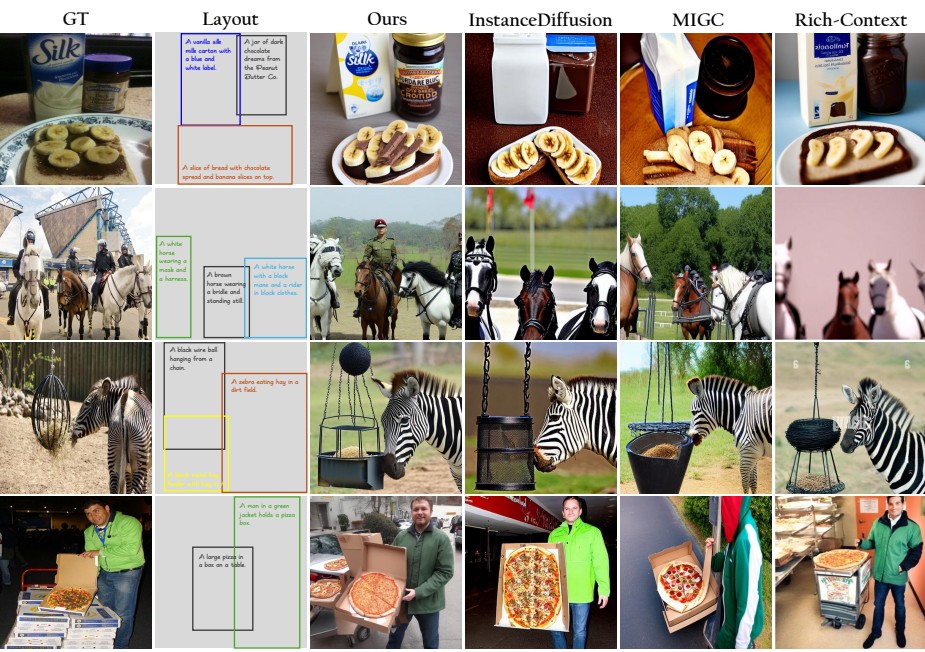

Figure 11: **Comparison results with ground truth.**

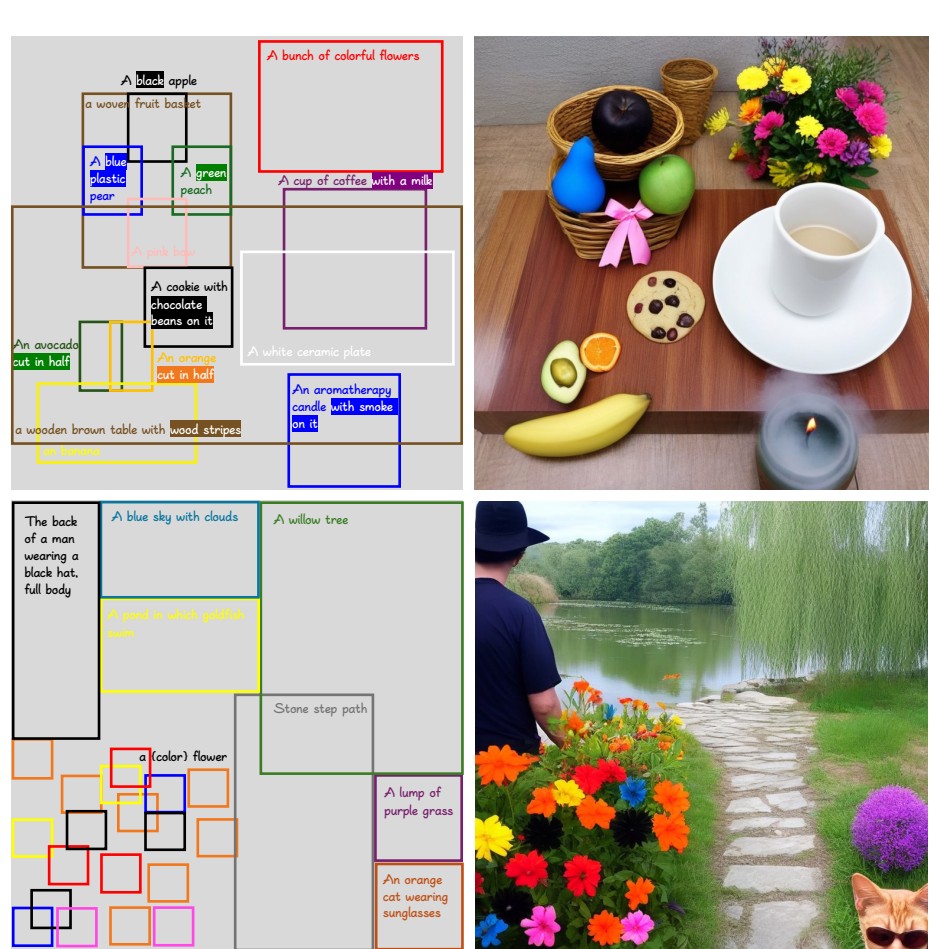

Figure 12: **Dense objects generation.**