# OpenReview forum: "IFAdapter: Instance feature control for grounded Text-to-Image Generation"
_ICLR.cc/2025/Conference — Submitted to ICLR 2025_

### Official Review · Reviewer_e5vG · 2024-10-24

**Soundness:** 3
**Presentation:** 3
**Contribution:** 3
**Rating:** 6
**Confidence:** 4

**Summary:**

The authors aim to tackle the challenge of achieving controllability in generating precise instance features. To do this, they introduce the Instance Feature Adapter (IFAdapter), designed for instance-level positioning and feature representation. Specifically, the IFAdapter employs learnable appearance queries that extract instance-specific feature information from descriptions, creating appearance tokens that complement EoT tokens. Additionally, the IFAdapter constructs a 2D semantic map to link instance features with designated spatial locations, providing enhanced spatial guidance. In areas where multiple instances overlap, a gated semantic fusion mechanism is utilized to mitigate feature confusion.

To validate their approach, the authors have created a new dataset, referred to as the COCO IFG benchmark. They leverage existing state-of-the-art Vision Language Models (VLMs) for annotation, resulting in a dataset with detailed instance-level descriptions. Experimental results indicate that the proposed plug-and-play component surpasses baseline models in both quantitative and qualitative assessments.

**Strengths:**

1. A plug-and-play component that can be integrated with various existing models to enhance layout control capabilities.
2. The paper also includes a user study, offering valuable insights into the proposed method.

**Weaknesses:**

1. The quality of the proposed dataset has not been evaluated. It appears that all ground truths (GTs) are generated by existing Vision Language Models (VLMs). A human-level quality assessment would be beneficial for greater impact within the community.
2. The assertion that the proposed component can “seamlessly empower various community models with layout control capabilities without retraining” (l.113) may be misleading. The IFAdapter is fundamentally a training-based method, and the phrase “without retraining” only holds true when applied to spaces closely aligned with COCO IFG, as the IFAdapter does not demonstrate zero-shot capabilities in this paper.
3. The semantic-instance map does not appear to be novel. Please refer to "BoxDiff: Text-to-Image Synthesis with Training-Free Box-Constrained Diffusion" (ICCV 2023) and other zero-shot L2I methods for comparison.
4. The appearance tokens show only minor improvements in Table 3. Additional explanations regarding this observation would be appreciated.

**Questions:**

Please kindly provide visual comparisons between the ground truth (GT) and the generated images.

---

> ### Author Response · Authors · 2024-11-20
>
> We thank the reviewer for their in-depth review and questions. We are pleased that they recognize the design philosophy behind our method and appreciate their acknowledgment of the valuable insights provided by the user study. Below are our point-by-point responses to the questions.
>
> > The quality of the proposed dataset has not been evaluated. It appears that all ground truths (GTs) are generated by existing Vision Language Models (VLMs). A human-level quality assessment would be beneficial for greater impact within the community.
>
> We believe that the cost of manually labeling a large-scale dataset is substantial. In order to ensure the quality of the labeling, we have chosen the state-of-the-art (SOTA) VLM and optimized the prompt engineering. To further assess the extent to which hallucinations affect our dataset, we conducted a user study. We manually reviewed 500 randomly selected samples from the training dataset, of which about 3% were identified as having location errors and 6% as having labeling hallucination errors, suggesting that the overall negative impact on model training is not particularly significant.
>
>
>
> > The assertion that the proposed component can “seamlessly empower various community models with layout control capabilities without retraining” (l.113) may be misleading. The IFAdapter is fundamentally a training-based method, and the phrase “without retraining” only holds true when applied to spaces closely aligned with COCO IFG, as the IFAdapter does not demonstrate zero-shot capabilities in this paper.
>
> We sincerely apologize for the misunderstanding, and we have revised our wording to eliminate the ambiguity.
> Our method is not limited to performing well only within the COCO IFG space, as all the cases shown in Fig. 1, Fig. 4, Fig. 9, and Fig. 10 are zero-shot results.
>
>
> > The semantic-instance map does not appear to be novel. Please refer to "BoxDiff: Text-to-Image Synthesis with Training-Free Box-Constrained Diffusion" (ICCV 2023) and other zero-shot L2I methods for comparison.
>
> We sincerely thank the reviewer for the valuable feedback. While the semantic-instance map approach is not proposed for the first time by us (as demonstrated in works like Spatext [1] and SSMG [2]), those methods fill the semantics of all instances into a single semantic map to guide generation. This can lead to semantic conflicts in regions where multiple instances overlap. In contrast, our method generates feature maps for each instance separately and fuses them into the final feature map using a gated semantic fusion mechanism, effectively resolving the semantic conflict issues might present in previous methods. We have analyzed this gated semantic fusion mechanism in detail in Appendix E.
>
> Regarding zero-shot methods, we compared our approach with DenseDiffusion and Multi-Diffusion. We also considered including BoxDiff as a baseline but ultimately decided against it for the following reasons:
> - BoxDiff requires concatenating all instance descriptions into the caption, and the total length of these descriptions cannot exceed the maximum input length for CLIP. However, in our instance-level dense captioning setup, there are cases where the concatenated descriptions exceed this limit.
> - In BoxDiff, each bounding box (bbox) typically corresponds to a single noun. In contrast, our benchmark associates an entire description with each instance, requiring a bbox to correspond to multiple words. Our experiments show that this mismatch causes BoxDiff to generate corrupted results.
> We provide some example cases at this [link](https://anonymous.4open.science/r/ICLR_rebuttal-762/vis_boxdiff.md)  to illustrate the issues.
>
>
> > The appearance tokens show only minor improvements in Table 3. Additional explanations regarding this observation would be appreciated.
>
> Thank you very much for your thoughtful suggestion. The appearance tokens improved the IFS rate by 10 points, which we believe is not insignificant.
>
> The relatively smaller improvement compared to the boost brought by EoT tokens can be attributed to the specific role of appearance tokens in assisting the generation of high-frequency details. Since the EoT token already encapsulates a certain degree of coarse appearance semantics, it is sufficient to effectively generate basic instance appearances in many cases. Appearance tokens, on the other hand, primarily aid in handling more challenging cases, which constitute a smaller proportion of all cases. Consequently, their overall contribution appears relatively modest.

---

> ### Author Response · Authors · 2024-11-20
>
> > Please kindly provide visual comparisons between the ground truth (GT) and the generated images.
>
> We thank the reviewer for their feedback. Since Fig. 1, Fig. 4, Fig. 9, and Fig. 10 all showcase zero-shot results, there are no corresponding ground truth (GT) images. However, we agree that it is important to provide comparisons that include GT results for reference. Therefore, we have added visual comparisons between the ground truth (GT) and the generated images, which can be accessed at this [link](https://anonymous.4open.science/r/ICLR_rebuttal-762/comparisons_with_GT.md) or Fig. 11 in the revised manuscript.
>
> ## References:
>
> [1] Avrahami, Omri, et al. "Spatext: Spatio-textual representation for controllable image generation." Proceedings of the IEEE/CVF Conference on Computer Vision and Pattern Recognition. 2023.
>
> [2] Jia, Chengyou, et al. "Ssmg: Spatial-semantic map guided diffusion model for free-form layout-to-image generation." Proceedings of the AAAI Conference on Artificial Intelligence. Vol. 38. No. 3. 2024.

---

> ### Author Response · Authors · 2024-12-01
>
> Dear Reviewer,
>
> As the rebuttal is coming to a close, we would like to kindly provide a gentle reminder that we have posted a response to your comments. If you don't mind, may we please check if our responses have addressed your concerns and improved your evaluation of our paper? We are happy to provide further clarifications to address any other concerns that you may still have before the end of the rebuttal.
>
> Sincerely,
>
> Authors

---

### Official Review · Reviewer_Nx7Q · 2024-11-03

**Soundness:** 2
**Presentation:** 3
**Contribution:** 2
**Rating:** 5
**Confidence:** 5

**Summary:**

- This work tackles instance Instance Feature Generation (IFG) task, i.e. train a model that can generate images given a global caption, spatial locations and detailed local captions as conditioning.
- Authors introduce IFAdapter, a plug-and-play module for improving IFG.
- The IFAdapter first extracts a fixed number of appearance tokens from a detailed instance caption and its spatial location. Next, a 2D map called Instance Semantic Map (ISM) is constructed using the bounding boxes of instances to aid the adherence of the model to spatial location conditions.
- IFAdapter is architecture agnostic and can be incorporated into existing open-source text to image models and authors show its effectiveness on the newly introduced IFG Benchmark constructed from the COCO dataset.

**Strengths:**

- The perceiver like Resampler design to extract fixed set of appearance tokens is novel and effective. This addresses the problem of only utilizing the EoT token from the text encoders.
- The gated semantic fusion to address multiple overlapping instances is very useful for layout to image generation methods.
- The proposed method is simple and is architecture agnostic.

**Weaknesses:**

- Authors claim IFG as their first contribution. But this task has already been introduced in InstanceDiffusion [Wang et. al 2024c]. InstanceDiffusion uses detailed instance captions in addition to the global caption. What is the difference between their setup and IFG?
- The experimental section needs heavy work to make this work ready for publication. With exponential progress in generative modeling, it is hard to control the experimental settings with other models, especially the training data. But that doesn't mean all other settings can be held constant to properly understand the contributions. InstanceDiffusion and GLIGEN use SD1.5 as their base generation model but authors use SDXL an already powerful base generator. This makes it hard to understand the improvements in Table 1 and 2. I recommend authors report numbers with SD1.5 as the base generator or retrain InstanceDiffusion with SDXL (since their code is available) to properly support their claims.
- Authors introduce a new benchmark and evaluation metric for this task. Why can't they use the evaluation setup and metrics as InstanceDiffusion? If authors find flaws in InstanceDiffusion's setup, I recommend authors point it out and discuss the advantages of the IFG Benchmark (setup) and IFS Rate (metric). There is no point in creating multiple new benchmarks when existing ones are already rigorous. For IFS Rate authors use Grounding DINO whereas InstanceDiffusion uses YOLO. Please compare with InstanceDiffusion using their exact setup (COCO and LVIS val set and their metrics) to support your claims.
- Authors claim that the a lightweight network $f$ provides an "importance" score for location (x,y) in the ISM construction and use it to compute $D(x,y)$. Please show qualitative or quantitative evidence that the network $f$ infact does what is claimed in the paper. While the idea sounds reasonable, I suspect how $f$ learns to predict the right "importance" scores without supervision.

**Questions:**

A few suggestions to improve the paper.
- Please remove point 3 in contributions. Comprehensive experiments are not a contribution but are rather required to support the claims made in the paper.
- The related works section has not been used correctly in this work according to my opinion. Authors just cite all relevant work but fail to differentiate their work from the literature. Please discuss how IFG is different from prior work and how their method is different from previously proposed methods in the related works section.
- If authors believe local CLIP score is suboptimal. I would recommend authors show (quantiatively) why VLMs are better than CLIP for this task. Please refrain from introducing a new metric and benchmark unless absolutely necessary.
- L77-78 is a repetition.

I'm willing to improve my rating if authors address the weakness section satisfactorily.

---

> ### Author Response · Authors · 2024-11-20
>
> We thank the reviewer for their careful review and toughtful questions. We are honored that they found our model design to be both simple and effective. Below are our point-by-point responses to the questions.
>
>
> >Authors claim IFG as their first contribution. But this task has already been introduced in InstanceDiffusion. InstanceDiffusion uses detailed instance captions in addition to the global caption. What is the difference between their setup and IFG?
>
> Thank you very much for your feedback. InstanceDiffusion is an impressive and pioneering work. Based on our understanding, the primary setup of InstanceDiffusion consists of three key aspects:
> - Allowing flexible conditioning (points, scribble, etc).
> - Enabling basic attribute (texture, color) binding for generated instances.
> - Supporting dense multi-instance generation.
>
> For attribute binding, the experiments in InstanceDiffusion primarily demonstrate its capability to bind single colors or textures. In the limitations section of InstanceDiffusion, the authors state: "We also find that texture binding for instances poses a challenge across all methods tested, including InstanceDiffusion." Additionally, through experiments (Fig. 1), we observed that more complex attribute binding problems remain unresolved. Therefore, we consider the InstanceDiffusion setup to be well-suited for generating accurate instance positions with multi-type conditioning and simple attribute binding.
>
> In contrast, our setup specifically targets the generation of complex instance-level attributes, such as textures, mixed colors, patterns, and more—problems that InstanceDiffusion has yet to solve. The appearance tokens and instance semantic maps proposed in our IFA are designed specifically to address these challenges in complex attribute binding. Furthermore, to evaluate the accuracy of complex instance-level attribute generation, we are the first to propose a verification pipeline based on Vision-Language Models (VLMs).

---

> > ### Author Response · Authors · 2024-11-20
> >
> > > The experimental section needs heavy work to make this work ready for publication. With exponential progress in generative modeling, it is hard to control the experimental settings with other models, especially the training data. But that doesn't mean all other settings can be held constant to properly understand the contributions. InstanceDiffusion and GLIGEN use SD1.5 as their base generation model but authors use SDXL an already powerful base generator. This makes it hard to understand the improvements in Table 1 and 2. I recommend authors report numbers with SD1.5 as the base generator or retrain InstanceDiffusion with SDXL (since their code is available) to properly support their claims.
> >
> > We thank the reviewer for the suggestion. In our understanding, Layout2Image (L2I) methods, such as GLIGEN, InstanceDiffusion, and IFA, fundamentally aim to control the base generator (e.g., by specifying instance positions and attributes). As such, the semantic generation capability of the base generator inherently determines the semantic limits of these methods.
> > To address the task of generating complex instance-level attributes, we believe that employing a generator with a broader semantic space, such as SDXL, is essential in practical applications. However, since previous methods have not been implemented on SDXL (which differs from SD 1.x in terms of architecture, making it a non-trivial task), we consider exploring how to incorporate L2I control into SDXL and successfully implement it as one of our contributions. The implementation details of our method (provided in the Appendix A) may offer insights for upgrading existing L2I methods to SDXL.
> >
> > We fully agree that comparisons based on the same generator can more directly demonstrate the effectiveness of our method. We have attempted to implement InstanceDiffusion XL; however, this has proven to be a challenging task. During the training process, we encountered several issues and raised them on InstanceDiffusion repository in hopes of receiving assistance from the original authors. Although we have not yet received a response, we will continue to monitor the situation closely.
> >
> > As a supplement, we report a comparison with Rich-Context [1], a concurrent work that shares a similar setup and is also trained on SDXL. We hope this demonstrates the improvements brought by our method. The experimental results are shown below:
> >
> > | Methods        | QwenVL | InternVL | CogVL | AP  | CLIP | FID |
> > |----------------|--------|----------|-------|-----|------|-----|
> > | IFA            | 79.7   | 68.6     | 61    | 49  | 25.1 | 22  |
> > | Rich-Context   | 74.4   | 52.7     | 37.8  | 40.6| 23.4 | 33  |
> >
> > We found that our method outperforms the Rich-Context. We attribute this improvement to our design philosophy: applying control only to a small subset of the most critical layers, thereby minimizing interference with the base generator's original generative capabilities. In contrast, the Rich-Context approach trains the U-Net, which may negatively impact the base model's generative performance.
> > We also provide qualitative comparisons, as shown in [This link](https://anonymous.4open.science/r/ICLR_rebuttal-762/Comparision_between_IFA_richcontext.md), whereas our method produces higher-quality images. This observation also explains why our approach achieves significantly better FID scores. We hope this comparison addresses your concerns.

---

> > > ### Comment · Reviewer_Nx7Q · 2024-11-23
> > > **Response to comments on backbone architectures**
> > >
> > > I agree that a powerful architecture is needed in real world to see the improvements. But for scientific study, it is important to control the variables to fully understand the results. I appreciate the reviewers trying to re-implement instancediffusion with SDXL. Can the authors instead train their method with SD1.5 and compare to instancediffusion? The comparison to Rich-Context is not as informative given the setup of instancediffusion is exactly the same as IFG.

---

> > > > ### Comment · Reviewer_Nx7Q · 2024-11-23
> > > > **Final comments on rebuttal**
> > > >
> > > > The authors have done a great job at addressing my concerns but two of my major concerns weren't answered satisfactorily.
> > > > 1. At an implementation level, what is the difference between InstanceDiffusion and IFG. Both the setups take location and instance level descriptions as condition to generate images. Please clarify the difference with examples. If evaluating IFG is the only contribution then authors should rewrite their introduction to reflect that the setup is not a contribution.
> > > > 2. Authors show that InstanceDiffusion is superior to their network on adhering to location conditions (From COCO evaluations on localization) and the marginal improvement on local CLIP score. But on their benchmark, they claim their method is superior. To rest all doubts, I recommend authors re-implement their work with a SD1.5 backbone and recompute numbers on both InstanceDiffusion setup and IFG Benchmark to solidfy their claims about performance.
> > > >
> > > > Since these two are important results, I cannot increase my scores at the moment.

---

> > > > > ### Author Response · Authors · 2024-11-23
> > > > >
> > > > > We sincerely appreciate the reviewer’s insightful follow-up comments. Below are our responses.
> > > > >
> > > > > > At an implementation level, what is the difference between InstanceDiffusion and IFG. Both the setups take location and instance level descriptions as condition to generate images. Please clarify the difference with examples. If evaluating IFG is the only contribution then authors should rewrite their introduction to reflect that the setup is not a contribution.
> > > > >
> > > > > In terms of input (instance-level description + location), we agree that our setup is similar to InstanceDiffusion. Following the suggestion, we have revised our abstract and introduction sections.
> > > > >
> > > > > However, it is worth noting that InstanceDiffusion’s evaluations did not focus on instance-level descriptions. In contrast, our setup introduces the IFG evaluation pipeline to assess the fidelity of instance-level feature generation. Moreover, IFAdapter has achieved the best performance results in both the IFG evaluation and user studies.
> > > > >
> > > > >
> > > > > > Table 2 in InstanceDiffusion computes local clip score to evaluate IFG. I would recommend authors to compute local clip score on the exact same data as InstanceDiffusion (or run instancediffusion on your data and compute the numbers; whichever is easiest) to compute the IFG capabilities of both the models.
> > > > >
> > > > > In response to the previous round of comments, we have provided the local-CLIP scores for the baselines on the COCO IFG benchmark. The results are as follows:
> > > > >
> > > > > | Methods            | dataset | IFA   | Instancediffusion | MIGC  | MultiDiffusion | DenseDiffusion | Gligen |
> > > > > |--------------------|---------|-------|-------------------|-------|----------------|----------------|--------|
> > > > > | Local Clip         | 21.31   | 21.09 | 20.83             | 19.64 | 19.92          | 20.52          | 18.73  |
> > > > >
> > > > > > I appreciate the authors trying to re-implement instancediffusion with SDXL. Can the authors instead train their method with SD1.5 and compare to instancediffusion? The comparison to Rich-Context is not as informative given the setup of instancediffusion is exactly the same as IFG.
> > > > >
> > > > > We sincerely thank the reviewer for their insightful suggestion. Our concurrent work, Rich-Context, shares a very similar IFG setup with IFAdapter and InstanceDiffusion, as all these methods utilize location and instance-level descriptions as conditional information. Additionally, Rich-Context includes comparative experiments with InstanceDiffusion (Table 1).
> > > > > Therefore, if the reviewer’s concern lies in the consistency of the setup, we believe the comparison between IFAdapter and Rich-Context is fair and provides valuable insights. We would greatly appreciate it if the reviewer could kindly elaborate on why they find this comparison not sufficiently informative, as this would help us better understand and address their concerns.
> > > > >
> > > > > As for training on SD1.5, it requires a significant amount of time. Given the limited rebuttal period, we cannot guarantee that the results will be ready before the rebuttal concludes. In theory, both the SD1.5 and SDXL versions of IFAdapter utilize instance feature maps and appearance tokens (whose effectiveness and rationale have been validated in our ablation studies). Therefore, their strengths and limitations should remain consistent.
> > > > >
> > > > > > Authors show that InstanceDiffusion is superior to their network on adhering to location conditions (From COCO evaluations on localization) and the marginal improvement on local CLIP score. But on their benchmark, they claim their method is superior.
> > > > >
> > > > > We believe the reasons for this difference may be as follows:
> > > > > - The IFG benchmark filters out small-sized objects (occupying less than 5% of the area) because such small objects tend to cause excessive hallucinations in VLMs. Given that IFAdapter does not introduce significant deviations in positional guidance for medium and large instances, it performs well in terms of position generation accuracy.
> > > > > - As shown in Fig. 1, InstanceDiffusion encounters semantic generation errors in complex cases (e.g., "Red deck chair with yellow star on it," where InstanceDiffusion fails to generate the chair). As a result, even though the positional generation is correct, the instance is counted as a wrong case due to semantic generation errors.
> > > > >
> > > > > We remain confident that our proposed method performs better in practical applications. It more effectively follows instructions to generate instance objects while maintaining positional accuracy, aligning closely with real-world scenarios. This has been further demonstrated through visualizations and user studies.

---

> > > > > ### Author Response · Authors · 2024-11-29
> > > > > **Response to request for experiments of IFA on SD1.5**
> > > > >
> > > > > > To rest all doubts, I recommend authors re-implement their work with a SD1.5 backbone and recompute numbers on both InstanceDiffusion setup and IFG Benchmark to solidfy their claims about performance.
> > > > >
> > > > > We sincerely appreciate the reviewers' valuable suggestions. Thanks to the extension of the rebuttal phase, we successfully completed experiments with the IFAdapter on SD15. The results of these experiments are presented below.
> > > > >
> > > > > Results under the InstanceDiffusion setup are as follows:
> > > > > | Methods            | $AP^{box}$ | $AP^{box}_{50}$ | $AR^{box}$ |
> > > > > |--------------------|------------|------------------|------------|
> > > > > | InstanceDiffusion  | 38.8       | 55.4             | 52.9       |
> > > > > | IFA (SDXL)         | 29.4       | 52.9             | 39.8       |
> > > > > | IFA  (SD1.5)       | 25.7       | 51.2             | 38.1       |
> > > > >
> > > > >
> > > > > Results under the IFG setup are as follows:
> > > > > | Methods        | QwenVL | InternVL | CogVL | AP  | CLIP | FID |
> > > > > |----------------|--------|----------|-------|-----|------|-----|
> > > > > | IFA(SDXL)           | 79.7   | 68.6     | 61.0    | 49  | 25.1 | 22  |
> > > > > | IFA(SD1.5)          | 73.8   | 55.3     | 43.1  | 46.1| 24.7 | 26.0  |
> > > > > | InstanceDiffusion   | 69.6   | 49.7     | 38.2  | 43.1| 23.3 | 26.8|
> > > > >
> > > > >
> > > > > From the above data, we can derive the following observations:
> > > > >
> > > > > - A stronger base model indeed improves the accuracy of instance feature generation.
> > > > > - Experimental results demonstrate that even within the SD1.5 architecture, IFAdapter outperforms InstanceDiffusion on the IFG task, further validating the effectiveness of IFAdapter.
> > > > > - Regarding location generation accuracy under the InstanceDiffusion setup, IFAdapter based on different base models shows no significant variation in metrics.
> > > > >
> > > > > The reason for the discrepancies in location generation accuracy (AP) across the two benchmarks has been explained in our previous comments:
> > > > >
> > > > > - The IFG benchmark filters out small-sized objects (occupying less than 5% of the area), as such objects tend to cause excessive hallucinations in vision-language models (VLMs). Since the IFAdapter does not introduce substantial deviations in positional guidance for medium and large instances, it maintains strong performance in terms of location generation accuracy.
> > > > > - As shown in Fig. 1, InstanceDiffusion encounters semantic generation errors in complex cases (e.g., "a red deck chair with a yellow star on it," where InstanceDiffusion fails to generate the chair). Consequently, even when positional generation is accurate, such instances are counted as errors due to semantic inaccuracies.
> > > > >
> > > > > We believe this explanation remains valid, as the IFAdapter based on different base models shows minimal changes in location generation accuracy under the InstanceDiffusion setup. The primary variation lies in feature generation accuracy under the IFG setup.
> > > > >
> > > > > We hope this comparative experiment addresses the reviewers' concerns.

---

> > > > > > ### Comment · Reviewer_Nx7Q · 2024-12-02
> > > > > > **Response to author rebuttal**
> > > > > >
> > > > > > I thank the authors for their additional experiments and detailed response.
> > > > > > 1. I appreciate the authors correcting their abstract and introduction to reflect that they are not introducing a new task but instead improving on a particular aspect of layout-to-image generation.
> > > > > > 2. To summarize, the contributions of this work are IFAdapter and a benchmark to evaluate the text adherence of a model to instance captions.
> > > > > > 3. I would like to politely disagree with the authors that their "method performs better in practical scenarios" since they removed a very practical capability (generating small objects) from their method. Moreover, removing objects that the evaluation setup cannot handle is not a good way to propose a new benchmark. Instead, I propose authors keep all kinds of objects and in addition report the ground-truth accuracy of the VLMs on different sized objects as a part of the evaluation so readers can get a complete picture on which numbers to trust. In its current form I believe the benchmark is not ready for publication. This weakens one significant contribution of the work. Hence, I vote to retain my rating.

---

> > > > > > > ### Author Response · Authors · 2024-12-03
> > > > > > >
> > > > > > > We sincerely appreciate the reviewer’s valuable feedback.
> > > > > > >
> > > > > > > 1. The reviewer suggested that our method might "remove the ability to generate small objects." We respectfully disagree with this viewpoint. While our method does exhibit certain deviations in the generation of extremely small objects, leading to a significant drop in AP for small instances under the InstanceDiffusion setup, this issue typically does not affect practical applications, since these deviations are relatively minor compared to th image size. Moreover, it does not imply that our method is incapable of generating small objects. On the contrary, as demonstrated in the visualization in Fig. 12, IFAdapter showcases strong layout control and attribute control capabilities for densely packed small objects.
> > > > > > >
> > > > > > > 2. The IFG benchmark is designed to verify the accuracy of instance feature generation. Complex features are more prominently observed in medium-to-large instances. Including small instances in the evaluation would result in these cases degenerating into close-set instance generation tasks, which could interfere with the benchmark's ability to effectively assess complex feature generation. Therefore, for evaluating instance feature generation, we believe our setup is more aligned with practical scenarios.

---

> > > > > ### Author Response · Authors · 2024-12-01
> > > > >
> > > > > Dear Reviewer,
> > > > >
> > > > > As the rebuttal is coming to a close, we would like to kindly provide a gentle reminder that we have posted a response to your comments. If you don't mind, may we please check if our responses have addressed your concerns and improved your evaluation of our paper? We are happy to provide further clarifications to address any other concerns that you may still have before the end of the rebuttal.
> > > > >
> > > > > Sincerely,
> > > > >
> > > > > Authors

---

> > ### Comment · Reviewer_Nx7Q · 2024-11-23
> > **Response to claims about 1st contribution.**
> >
> > I thank the authors for their detailed response.
> > I agree that instancediffusion has other contributions in addition to detailed instance generation. Getting down to the details, InstanceDiffusion's setup differs from GLIGEN in one very important aspect. GLIGEN uses semantic category along with location information as conditioning. InstanceDiffusion uses very detailed instance level captions (in addition to global caption) along with grounding information. InstanceDiffusion might not perform well with texture binding but the setup of InstanceDiffusion involves using grounding information along with detailed instance captions which is the same as this current work. I request the authors to explain to me at a benchmark level how this differs from the one proposed by IFG? Please feel free to use examples to describe the differences. I'm still not convinced the proposed task is novel given InstanceDiffusion has already proposed to do this.

---

> ### Author Response · Authors · 2024-11-20
>
> > Authors introduce a new benchmark and evaluation metric for this task. Why can't they use the evaluation setup and metrics as InstanceDiffusion? If authors find flaws in InstanceDiffusion's setup, I recommend authors point it out and discuss the advantages of the IFG Benchmark (setup) and IFS Rate (metric). There is no point in creating multiple new benchmarks when existing ones are already rigorous. For IFS Rate authors use Grounding DINO whereas InstanceDiffusion uses YOLO. Please compare with InstanceDiffusion using their exact setup (COCO and LVIS val set and their metrics) to support your claims.
>
> We agree that the experimental setup of InstanceDiffusion is comprehensive and rigorous. However, the main experiments of InstanceDiffusion (Table 1) primarily focus on evaluation metrics for object positions and do not include metrics for assessing instance features, which is the core focus of our setup. Therefore, our IFG Benchmark is designed not only to evaluate positional accuracy (AP) but also to complementarily assess the quality of instance feature generation using the IFS Rate.
>
> We chose to use Grounding DINO because it has been pre-trained on large-scale datasets, offering higher generalization capabilities compared to YOLO. Given that our setup targets the generation of instances with complex features, we believe Grounding DINO is a more suitable choice.
>
> Based on the reviewer's suggestion, we have reported the performance of our method on the COCO validation set as a reference. We did not include results on the LVIS dataset because InstanceDiffusion does not provide validation scripts for the LVIS dataset.
>
> | Methods            | $AP^{box}$ | $AP^{box}_{50}$ | $AR^{box}$ | $AP^{box}_{L}$ | $AP^{box}_{s}$ |
> |--------------------|------------|------------------|------------|----------------|----------------|
> | InstanceDiffusion  | 38.8       | 55.4             | 52.9       | -              | -              |
> | IFA                | 29.4       | 52.9             | 39.8       | 42.0           | 6.0            |
>
> The experimental results suggest that our method performs worse than InstanceDiffusion under its experimental setup. As shown in the table, our method achieves lower $AP^{box}_{s}$ for small objects. After visualizing the samples, we believe this is primarily due to the presence of many small instances in the COCO validation set. Our approach leverages an Instance Semantic Map with a 16×16 resolution as guidance. Consequently, for bounding boxes occupying small areas, positional deviations are relatively large, leading to inaccuracies in the position generation of small instances. In contrast, methods like InstanceDiffusion, which represent objects using tokens, are relatively better for small-object generation.We appreciate the reviewer for pointing out this limitation of our method, and we will include this shortcoming in the limitations section of our paper.
>
> But we would like to highlight that this shortcoming does not impact IFA's performance under our setup. This is because only objects with larger areas typically exhibit noticeable appearance features, and generating small objects is not the primary target of the IFG task.
>
> > Authors claim that the a lightweight network $f$ provides an "importance" score for location (x,y) in the ISM construction and use it to compute D(x, y). Please show qualitative or quantitative evidence that the network f infact does what is claimed in the paper. While the idea sounds reasonable, I suspect how f learns to predict the right "importance" scores without supervision.
>
> We thank the reviewer for their suggestion. We have added Appendix E to discuss the relevant content, and the importance score visualization can be found [there](https://anonymous.4open.science/r/ICLR_rebuttal-762/visual_importance_score.md). Here, we provide a brief summary of the discussion.
>
> Despite the lack of explicit supervision, 𝑓 implicitly learns from the dataset through the denoising loss to identify salient features in the Instance Semantic Map at various positions and assign greater weights to the corresponding instances. We hypothesize that this may be due to diffusion model features inherently containing a certain degree of depth priors, which 𝑓 learns to map to weights during the training process.
>
> > Please remove point 3 in contributions. Comprehensive experiments are not a contribution but are rather required to support the claims made in the paper.
>
> We thank the reviewer for the feedback. In response to the identified weaknesses, we have added the corresponding experiments to strengthen our claims. We hope this addresses your concerns.

---

> > ### Comment · Reviewer_Nx7Q · 2024-11-23
> > **Response to author's comments on evaluation.**
> >
> > Looking at the results on localization (only AR i.e. recall) tells me that the proposed method is not as good as InstanceDiffusion in following all the grounding information. I agree that the focus of the work is to evaluate IFG but at the end of the day, the methods need to be evaluated on their capabilities to follow grounding information. So this information is equally crucial for the work. InstanceDiffusion, despite using a weaker backbone adheres better to grounding information.
> >
> > Table 2 in InstanceDiffusion computes local clip score to evaluate IFG. I would recommend authors to compute local clip score on the exact same data as InstanceDiffusion (or run instancediffusion on your data and compute the numbers; whichever is easiest) to compute the IFG capabilities of both the models. Without this, it is not possible to correctly judge this work. You can divide the area of bounding boxes into small, medium, large an compute local clip score (not with the whole image but within a bounding box and its caption) within each on COCO-val (similar to instanceDiffusion) for both the methods.
> >
> > I agree with the authors that using GroundingDINO is better for such tasks.
> > I thank the reviewer for the visualization. It is surprising to me that the model learns this information in an unsupervised manner! Nice finding!

---

> ### Author Response · Authors · 2024-11-20
>
> >The related works section has not been used correctly in this work according to my opinion. Authors just cite all relevant work but fail to differentiate their work from the literature. Please discuss how IFG is different from prior work and how their method is different from previously proposed methods in the related works section.
>
> We thank the reviewer for their suggestion. We have revised the Related Work section accordingly in the newly uploaded manuscript.
>
> >If authors believe local CLIP score is suboptimal. I would recommend authors show (quantiatively) why VLMs are better than CLIP for this task. Please refrain from introducing a new metric and benchmark unless absolutely necessary.
>
> Thanks very much for the reviewer’s suggestion. There are several reasons why we do not use CLIP score as an evaluation metric:
> - CLIP score measures the alignment between the global feature vectors of an image and a text description. However, previous works [2, 3] have pointed out that CLIP tends to treat sentence features as a "bag of words," making it difficult to distinguish fine-grained details. For example, CLIP struggles to differentiate between "a person wearing a white shirt and black pants" and "a person wearing black pants and a white shirt." This limitation means that CLIP score cannot effectively evaluate the correctness of feature binding. In contrast, Vision-Language Models (VLMs) encode images at the patch level, offering finer-grained perception. Additionally, our prompt engineering guides VLMs to focus more on detailed features, making them more effective for detecting instance-level attributes.
> - When calculating evaluation metrics, it is generally desirable for the metric to have a clear upper bound, typically represented by the performance on ground truth (real images). However, as observed in Table 2 of MIGC [4] paper, CLIP scores for some methods exceed those of real images. This makes it difficult to interpret the meaning of the CLIP score values.
> - In contrast, our IFS Rate can be interpreted as the generation success rate, offering a more intuitive understanding compared to CLIP score.
>
> As a reference, we also report experimental results using local CLIP score in the table below. The results show that our method achieves competitive performance on this metric, closely approaching the scores obtained on the dataset.
>
> | Methods            | IFA   | Instancediffusion | MIGC  | MultiDiffusion | DenseDiffusion | Gligen | dataset |
> |--------------------|-------|-------------------|-------|----------------|----------------|--------|---------|
> | Local Clip         | 21.09 | 20.83             | 19.64 | 19.92          | 20.52          | 18.73  | 21.31   |
>
> > L77-78 is a repetition.
>
> We thank the reviewer for the careful review. The issue has been addressed.
>
> ## References:
> [1] Cheng, Jiaxin, et al. "Rethinking The Training And Evaluation of Rich-Context Layout-to-Image Generation." The Thirty-eighth Annual Conference on Neural Information Processing Systems.
>
> [2] Yuksekgonul, Mert, et al. "When and why vision-language models behave like bags-of-words, and what to do about it?." arXiv preprint arXiv:2210.01936 (2022).
>
> [3] Wu, Yinwei, Xingyi Yang, and Xinchao Wang. "Relation Rectification in Diffusion Model." Proceedings of the IEEE/CVF Conference on Computer Vision and Pattern Recognition. 2024.
>
> [4] Zhou, Dewei, et al. "Migc: Multi-instance generation controller for text-to-image synthesis." Proceedings of the IEEE/CVF Conference on Computer Vision and Pattern Recognition. 2024.

---

> > ### Comment · Reviewer_Nx7Q · 2024-11-23
> > **Response to authors comment**
> >
> > - On what data was the local CLIP score computed here?
> > I thank the reviewer for correcting related works and typos.

---

### Official Review · Reviewer_V57v · 2024-11-03

**Soundness:** 4
**Presentation:** 3
**Contribution:** 3
**Rating:** 6
**Confidence:** 4

**Summary:**

This paper presents Instance Feature Adapter (IFA) for layout-to-image generation. The key insight of IFA is to incorporate additional appearance tokens corresponded to different objects, so as to steer the T2I models to generate images that convey precise layout information and text semantics. To achieve this, IFA first leverages learnable instance tokens to aggregate the textual and bounding box information of the specific objects. To cope with the feature leakage problems, IFA further introduce Instance Semantic Map strategy to reallocate the spatial area for different semantics, so as to alleviate the feature conflicts between different objects during external feature tokens injection process. A new benchmark is proposed, the visual improvement over different baselines is significant. Further, the proposed method is a plug-and-play module, which can be adapted to various community models.

**Strengths:**

1. The paper is well written and easy to follow.
2. The designed approaches efficiently incorporate the external object semantics and layout information into the generation process. The proposed Appearance Tokens aggregate the textual semantic and bbox information with learnable tokens. The proposed Instance Semantic Map accurately reallocates the spatial area of different objects, and solves the semantic fusion and feature leakage problem.
3. The illustrated visual results are impressive, which shows clear superiority against competing baselines.
4. The proposed method is compatible with various community models.

**Weaknesses:**

1. The design module is supposed to interface with the text encoders, and both Appearance Tokens and Instance Semantic Map introduce the attention mechanism. Will it be computational costly during the inference process. There should be a detailed discussion.
2. The size and shape of different objects seem unstable when applying IFA to different community models (Fig. 4), is it caused by the re-weight strategy from Instance Semantic Maps?
3. The L2I problem is not a novel task, and the main novelty mainly lies in the implementation detail of the layout incorporation strategy, which may not bring significantly inspirations to the community.

**Questions:**

Is the proposed model able to generate images with some intricate prompts. For example, a blue dog with red legs running on a colorful river, (with dog, legs, river assigned to different boxes). I want to see some limitations of the proposed method, or in other words, I want to know how IFA copes with the semantic issues which may inherit from the base model given out-of-domain prompts and instructions. I would love to revise my rating after further discussion with the authors.

---

> ### Author Response · Authors · 2024-11-20
>
> We thank the reviewer for the extensive review and insightful questions. We are glad they found our paper clear and appreciated the effectiveness of our approaches and also value the recognition of our method's compatibility with community models. Below are our point-by-point responses to the questions.
>
>
> > The design module is supposed to interface with the text encoders, and both Appearance Tokens and Instance Semantic Map introduce the attention mechanism. Will it be computational costly during the inference process. There should be a detailed discussion.
>
> We thank you for the suggestion. We have added a dedicated section (Appendix. F) to conduct experiments and discuss the computational cost of our method. The main results are summarized below. The findings indicate that, compared to the base model, IFA does not introduce significant computational overhead. We attribute this to the fact that IFAdapter applies control to only a small subset of cross-attention layers, making it relatively lightweight.
>
> | Methods                 | GPU Memory Usage (MiB) | Inference Latency (Seconds) |
> |-------------------------|-------------------------|------------------------------|
> | SDXL                    | 16601                   | 5.325966                     |
> | IFA (4 instances)       | 17875                   | 8.030823                     |
> | IFA (8 instances)       | 19485                   | 8.771261                     |
> | IFA (16 instances)      | 19651                   | 10.731308                    |
>
>
> >The size and shape of different objects seem unstable when applying IFA to different community models (Fig. 4), is it caused by the re-weight strategy from Instance Semantic Maps?
>
> We thank the reviewer for the feedback. We believe this issue is not caused by the re-weighting strategy, as it does not influence the shape of objects. Instead, we attribute it to two factors:
> - The BBox serves as a relatively flexible condition, primarily controlling object placement, which allows for the generation of objects with varying sizes and shapes. If specific object shapes are desired, they can be explicitly constrained in the instance description.
> - Different community models are fine-tuned on various datasets, which may result in differing preferences for object shapes. A clear example is that LEGO-style community models tend to generate more geometrically regular instances, whereas clay-style models are inclined to produce thinner objects.
>
>
> > The L2I problem is not a novel task, and the main novelty mainly lies in the implementation detail of the layout incorporation strategy, which may not bring significantly inspirations to the community.
>
> We fully agree with the reviewer's perspective. Layouts to images (L2I) generation based solely on category labels (e.g., a "dog") is not a new problem. Therefore, our IFG task represents a rethinking and enhancement of the L2I task. In our task, the focus is not only on ensuring the accurate generation of object categories and positions but also on guaranteeing the fidelity of each object's appearance (e.g., a "dog wearing a swimsuit"). This makes the task more challenging and practical. For instance, in generating indoor layout images, we often want to specify not just the position of furniture but also its shape, color, material, and more. Such scenarios fall outside the scope of typical L2I methods but are effectively addressed by our proposed IFA.
>
> Additionally, beyond introducing a layout incorporation strategy, our method represents the initial attempt to adapt L2I techniques to SDXL. This provides valuable insights for the community on how to upgrade existing L2I models to SDXL. Moreover, our method's ability to empower the design of various community models offers a novel perspective that can serve as a reference for future developments.

---

> ### Author Response · Authors · 2024-11-20
>
> > Is the proposed model able to generate images with some intricate prompts. For example, a blue dog with red legs running on a colorful river, (with dog, legs, river assigned to different boxes). I want to see some limitations of the proposed method, or in other words, I want to know how IFA copes with the semantic issues which may inherit from the base model given out-of-domain prompts and instructions.
>
> The primary role of IFA is to exert control over the generation of instance appearance and position; it is not designed to address out-of-domain  semantic issues. As such, IFA does not enhance the model's ability to generate content for prompts that are outside the semantic understanding capabilities of the base model. In other words, the semantic comprehension of prompts remains limited to that of the base model, and any semantic issues inherent to the base model will not be resolved by introducing IFA.
>
> For example, in the case of "a blue dog with red legs running on a colorful river," IFA can generate the image relatively accurately because the prompt does not completely deviate from the base model's generation space. However, for another typical out-of-distribution (OOD) case, such as "a horse rides the astronaut," where the horse and astronaut are assigned to different bounding boxes, IFA fails to produce a satisfactory image. If you are interested, we have showcased the image generation results for these two examples at [OOD cases](https://anonymous.4open.science/r/ICLR_rebuttal-762/OOD_case.md).

---

> ### Comment · Reviewer_V57v · 2024-11-23
>
> Thanks the authors for their detailed responses. Most of my concerns are addressed, and I decide to keep my original score.

---

### Official Review · Reviewer_mFKu · 2024-11-04

**Soundness:** 3
**Presentation:** 3
**Contribution:** 3
**Rating:** 6
**Confidence:** 5

**Summary:**

This paper aims to improve the capability of Text-to-Image diffusion models in generating precise features and positioning multiple instances in images. The proposed IFAdapter enhances feature accuracy by integrating additional appearance tokens and constructing an instance semantic map, ensuring that each instance's features align accurately with its spatial location. The IFAdapter is designed as a flexible, plug-and-play module, allowing enhanced control without retraining.

**Strengths:**

+ The authors propose an important task, namely Instance Feature Generation. In addition, the authors also provide a benchmark and a verification pipeline
+ The proposed method seems to achieve good results. Qualitative and quantitative experiments demonstrate the effectiveness of the proposed method.

**Weaknesses:**

-	Compared with instance diffusion and dense diffusion, this paper shows a small number of objects and does not show the situation where the object is denser.
-	The details of the method are not explained clearly. The details of the dataset construction and the baseline setting are not presented clearly. Ablation experiments lack a basic baseline presentation.
-	Authors should carefully check for errors in the text. For example, a sentence appears twice in succession in Introduction.

**Questions:**

1. Why IFAdapter enables it to be seamlessly applied to various community models. For example, appearance queries are trained , how do the authors ensure that the feature has sufficient generalization capabilities.
2. Why can appearance-related features be extracted from the bounding box through Fourier and MLP, especially when the model inference cannot obtain image input?
3. Is the model-free method based on the same backbone as the proposed method? Are all the comparison methods that require training trained on the provided dataset? The VLM used to annotate the proposed dataset is the same as the VLM used in the evaluation metric, which may be unfair to methods that are not trained on the proposed dataset.
4. VLMs are likely to produce hallucinations. How do the authors ensure that the annotations of the provided dataset are free of hallucinations?

---

> ### Author Response · Authors · 2024-11-20
>
> We thank the reviewer for the extensive review and insightful questions. We are pleased to note the importance of the Instance Feature Generation task and appreciate the recognition of our benchmark, verification pipeline, and the effectiveness of our method in both qualitative and quantitative experiments. Below are our point-by-point responses to the questions.
>
> > Compared with instance diffusion and dense diffusion, this paper shows a small number of objects and does not show the situation where the object is denser.
>
> We believe that the vast majority of scenes contain less than 10 instances, but our method is also able to generate images with more dense instances while maintaining consistency of detail. As shown in this [link](https://anonymous.4open.science/r/ICLR_rebuttal-762/Dense_objects_generation.md) or Fig. 12 in the revised manuscript, the flowers strictly follows the color and position of the instruction, while other details such as a black hat, a goldfish in a pond, and a cat's sunglasses are generated accurately, although we do not assign additional bounding boxes to them. In another indoor case, we also used a larger number of instances, and again, the fine-grained details of the instances were well generated (e.g., cutted avocados and oranges, smoke from candles), and we were pleasantly surprised to find that our method was also well generated on some counterintuitive objects (e.g., plastic-like blue pears, green fluffy peaches, black apple, etc.)
>
> > The details of the method are not explained clearly. The details of the dataset construction and the baseline setting are not presented clearly. Ablation experiments lack a basic baseline presentation.
>
> We have corrected several  errors in the paper, including those in Equation 5 and its corresponding description. But as Reviewer V57v mentioned that our method is easy to follow, we are unsure which parts might require clarification. If you could point out specific sections where our explanation is unclear, we would be very grateful.
> Regarding the dataset construction process, we have further refined the description and updated it in Appendix A. For the baseline settings, we have included detailed explanations in Appendix B. Additionally, we have conducted ablation experiments on the baselines and provided the corresponding analysis, you can see Table 3 in the revised manuscript.
>
> > Authors should carefully check for errors in the text. For example, a sentence appears twice in succession in Introduction.
>
> We thank the reviewer for their careful review. The issue has been addressed.
>
> > Why IFAdapter enables it to be seamlessly applied to various community models. For example, appearance queries are trained , how do the authors ensure that the feature has sufficient generalization capabilities.
>
> The IFAdapter is a lightweight plugin designed to control the generation process of the base model. For instance, appearance queries are trained to extract semantic information that benefits high-frequency details in instance generation, which may exist in a specific subspace of the text embedding.
> Community models, on the other hand, are stylized versions of the base model obtained by fine-tuning a small number of parameters on different datasets. These models typically share the similar feature space as the base model. Consequently, the control learned by the IFAdapter on the base model can be effectively transferred to community models.
>
> > Why can appearance-related features be extracted from the bounding box through Fourier and MLP, especially when the model inference cannot obtain image input?
>
> Appearance-related features are not extracted from the image; instead, they are derived from text embeddings through a resampler. These features capture semantics beneficial for generating high-frequency details, complementing the coarse instance semantics encoded in the EoT token. Since these features are used for appearance generation, we refer to them as appearance tokens.
> In Fig. 2, the green box labeled "BBox" represents a vector containing only [x1,y1,x2,y2]. The Resampler, on the other hand, serves as the primary structure for extracting appearance semantics from sentence features.

---

> > ### Author Response · Authors · 2024-11-20
> >
> > > Is the model-free method based on the same backbone as the proposed method? Are all the comparison methods that require training trained on the provided dataset? The VLM used to annotate the proposed dataset is the same as the VLM used in the evaluation metric, which may be unfair to methods that are not trained on the proposed dataset.
> >
> > All the models are based on the Stable Diffusion architecture and we used the official checkpoints and code provided by these baselines for comparison.
> > We acknowledge that using the same VLM for both annotation and evaluation may introduce a certain degree of bias. To address this, we employed three different VLMs for a more objective evaluation. Additionally, we conducted a user study, which is the most objective evaluation metric. We believe these efforts help ensure a fair comparison to a reasonable extent.
> >
> >
> > > VLMs are likely to produce hallucinations. How do the authors ensure that the annotations of the provided dataset are free of hallucinations?
> >
> > We cannot guarantee that the dataset is entirely free of hallucinations. To further assess the extent to which hallucinations affect our dataset, we conducted a user study. We manually reviewed 500 randomly selected samples from the training dataset, of which about 3% were identified as having location errors, and 6% were identified as having labeling hallucination errors, suggesting that the overall negative impact on model training is not particularly significant.

---

> ### Author Response · Authors · 2024-12-01
>
> Dear Reviewer,
>
> As the rebuttal is coming to a close, we would like to kindly provide a gentle reminder that we have posted a response to your comments. If you don't mind, may we please check if our responses have addressed your concerns and improved your evaluation of our paper? We are happy to provide further clarifications to address any other concerns that you may still have before the end of the rebuttal.
>
> Sincerely,
>
> Authors

---

### Meta-Review · Area_Chair_GWmU · 2024-12-22

**Metareview:**

Summary
This paper studies the problem of controllable image generation where the scene layout can be specified using a global caption, instance locations, and instance specific captions. The authors propose an adapter based technique called IFA that fuses such instance conditioning with a pretrained text to image backbone. The authors propose a benchmark, IFG, to evaluate their work.

Strengths
1. IFA is architecture agnostic which makes it generally and widely applicable to many text to image models.
2. The authors address a critical limitation of current methods that only use the last token from the text encoders to condition the image generation.
3. The method section in the paper is written well and explains the approach

Weaknesses
1. The stated contributions in the paper are inaccurate and required revisions after review. There are also factually vague statements (L143 in reference to conditioning on instance prompts which InstanceDiffusion does). Not having a clear sense of contributions in the paper is not acceptable.
2. The authors do not state any evidence for why IFG is a necessary benchmark compared to prior instance level generation benchmarks -- different metric, different VLM from prior work. They only talked about this difference in the rebuttal and not in the main paper. Given that this is a major contribution and the setting used for evaluating against all prior work, I believe this is a serious omission. The author rebuttal also does not answer this question raised by Nx7Q.
3. Comparisons in the main paper used a different text to image backbone across methods. This made it hard to make a fair comparison and assessment of improvements, if any. The authors did fix this in the rebuttal comments.

Justification
The paper's method section is well written. However, the exact contributions, relation to related work, reason for benchmark, experimental setup need revision (weaknesses above) and thus, it is better that this paper be resubmitted and go through the review process again.

**Additional Comments On Reviewer Discussion:**

Two of the reviewers did not engage with the authors despite the authors nudging them and gave a weaker acceptance rating.
Reviewer Nx7Q raised several concerns about the paper's writing, stated contributions, experimental protocol, value of benchmark, and necessity of change.
The authors replied to address these concerns, however, Nx7Q remains unconvinced, especially about the value of the benchmark in this work and the necessity to depart from prior benchmarks.

---

### Decision · Program_Chairs · 2025-01-22

Reject